# ON BONUS-BASED EXPLORATION METHODS IN THE ARCADE LEARNING ENVIRONMENT

**Adrien Ali Taïga**
MILA, Université de Montréal
Google Brain
`adrien.ali.taiga@umontreal.ca`

**William Fedus**
MILA, Université de Montréal
Google Brain
`liamfedus@google.com`

**Marlos C. Machado**
Google Brain
`marlosm@google.com`

**Aaron Courville** *
MILA, Université de Montréal
`aaron.courville@umontreal.ca`

**Marc G. Bellemare** *
Google Brain
`bellemare@google.com`

## ABSTRACT

Research on exploration in reinforcement learning, as applied to Atari 2600 game-playing, has emphasized tackling difficult exploration problems such as MON-TEZUMA'S REVENGE (Bellemare et al., 2016). Recently, bonus-based exploration methods, which explore by augmenting the environment reward, have reached above-human average performance on such domains. In this paper we reassess popular bonus-based exploration methods within a common evaluation framework. We combine Rainbow (Hessel et al., 2018) with different exploration bonuses and evaluate its performance on MONTEZUMA'S REVENGE, Bellemare et al.'s set of hard of exploration games with sparse rewards, and the whole Atari 2600 suite. We find that while exploration bonuses lead to higher score on MONTEZUMA'S REVENGE they do not provide meaningful gains over the simpler $\epsilon$-greedy scheme. In fact, we find that methods that perform best on that game often underperform $\epsilon$-greedy on easy exploration Atari 2600 games. We find that our conclusions remain valid even when hyperparameters are tuned for these easy-exploration games. Finally, we find that none of the methods surveyed benefit from additional training samples (1 billion frames, versus Rainbow's 200 million) on Bellemare et al.'s hard exploration games. Our results suggest that recent gains in MONTEZUMA'S REVENGE may be better attributed to architecture change, rather than better exploration schemes; and that the real pace of progress in exploration research for Atari 2600 games may have been obfuscated by good results on a single domain.

## 1 INTRODUCTION

In reinforcement learning, the exploration-exploitation trade-off describes an agent's need to balance maximizing its cumulative rewards and improving its knowledge of the environment. While many practitioners still rely on simple exploration strategies such as the $\epsilon$-greedy scheme, in recent years a rich body of work has emerged for efficient exploration in deep reinforcement learning. One of the most successful approaches to exploration in deep reinforcement learning is to provide an exploration bonus based on the relative novelty of the state. This bonus may be computed, for example, from approximate counts (Bellemare et al., 2016; Ostrovski et al., 2017; Tang et al., 2017; Machado et al., 2018a), from the prediction error of a dynamics model (ICM, Pathak et al., 2017) or by measuring the discrepancy to a random network (RND, Burda et al., 2019).

---

*CIFAR Fellow

Bellemare et al. (2016) argued for the importance of the Atari 2600 game MONTEZUMA'S REVENGE as a challenging domain for exploration. MONTEZUMA'S REVENGE offers a sparse reinforcement signal and relatively open domain that distinguish it from other Atari 2600 games supported by the Arcade Learning Environment (ALE; Bellemare et al., 2013). As a consequence, recent exploration research (bonus-based or not) has aspired to improve performance on this particular game, ranging from a much deeper exploration (6600 points; Bellemare et al., 2016), to completing the first level (14000 points; Pohlen et al., 2018; Burda et al., 2019), to super-human performance (400,000 points; Ecoffet et al., 2019).

Yet the literature on exploration in reinforcement learning still lacks a systematic comparison between existing methods, despite recent entreaties for better practices to yield reproducible research (Henderson et al., 2018; Machado et al., 2018b). One of the original tenets of the ALE is that agent evaluation should take on the entire suite of 60 available Atari 2600 games. Even within the context of bonus-based exploration, MONTEZUMA'S REVENGE is but one of seven hard exploration, sparse rewards games (Bellemare et al., 2016). Comparisons have been made between agents trained under different regimes: using different learning algorithms and varying numbers of training frames, with or without reset, and with and without controller noise (e.g. *sticky actions* Machado et al., 2018b). As a result, it is often unclear if the claimed performance improvements are due to the exploration method or other architectural choices, and whether these improvements carry over to other domains. The main conclusion of our research is that an over-emphasis on one Atari 2600 game, combined with different training conditions, may have obfuscated the real pace of progress in exploration research.

To come to this conclusion, we revisit popular bonus-based exploration methods (CTS-counts, PixelCNN-counts, RND, and ICM) in the context of a common evaluation framework. We apply the Rainbow (Hessel et al., 2018) agent in turn to MONTEZUMA'S REVENGE, Bellemare et al.'s seven hardest Atari 2600 games for exploration, and the full suite from the ALE.

**Source of exploration bonus.** We find that all agents perform better than an $\epsilon$-greedy baseline, confirming that exploration bonuses do provide meaningful gains. However, we also find that more recent exploration bonuses do not, by themselves, lead to higher performance than the older pseudo-counts method. Across the remainder of hard exploration games, we find that all bonuses lead to performance that is comparable, and in one case worse, than $\epsilon$-greedy.

**Performance across full Atari 2600 suite.** One may expect a good exploration algorithm to handle the exploration/exploitation trade-off efficiently: exploring in difficult games, without losing too much in games where exploration is unnecessary. We find that performance on MONTEZUMA'S REVENGE is in fact *anticorrelated* with performance across the larger suite. Of the methods surveyed, the only one to demonstrate better performance across the ALE is the non-bonus-based Noisy Networks (Fortunato et al., 2018), which provide as an additional point of comparison. Noisy Networks perform worse on MONTEZUMA'S REVENGE.

**Hyperparameter tuning procedure.** The standard practice in exploration research has been to tune hyperparameters on MONTEZUMA'S REVENGE then evaluate on other Atari 2600 games. By virtue of this game's particular characteristics, this approach to hyperparameter tuning may unnecessarily increase the behaviour of exploration strategies towards a more exploratory behavior, and explain our observation that these strategies perform poorly in easier games. However, we find that tuning hyperparameters on the original ALE training set does not improve performance across Atari 2600 games beyond that of $\epsilon$-greedy .

**Amount of training data.** By design, exploration methods should be sample-efficient. However, some of the most impressive gains in exploration on MONTEZUMA'S REVENGE have made use of significantly more data (1.97 billion Atari 2600 frames for RND, versus 100 million frames for the pseudo-count method). We find that, within our common evaluation framework, additional data does not play an important role on exploration performance.

Altogether, our results suggests that more research is needed to make bonus-based exploration robust and reliable, and serve as a reminder of the pitfalls of developing and evaluating methods primarily on a single domain.

## 1.1 RELATED WORK

Closest to our work, Burda et al. (2018) benchmark various exploration bonuses based on prediction error (Schmidhuber, 1991; Pathak et al., 2017) within a set of simulated environment including many Atari 2600 games. Their study differs from ours as their setting ignores the environment reward and instead learns exclusively from the intrinsic reward signal. Outside of the ALE, Osband et al. (2019) recently provide a collection of experiments that investigate core capabilities of exploration methods.

## 2 EXPLORATION METHODS

We focus on bonus-based methods, that is, methods that promote exploration through a reward signal. An agent is trained with the reward $r = r^{\text{ext}} + \beta \cdot r^{\text{int}}$ where $r^{\text{ext}}$ is the extrinsic reward provided by the environment, $r^{\text{int}}$ the intrinsic reward computed by agent, and $\beta > 0$ is a scaling parameter. We now summarize the methods we evaluate to compute the intrinsic reward $r^{\text{int}}$.

### 2.1 PSEUDO-COUNTS

Pseudo-counts (Bellemare et al., 2016; Ostrovski et al., 2017) were proposed as way to estimate counts in high dimension states spaces using a density model. The agent is then encouraged to visit states with a low visit count. Let $\rho$ be a density model over the state space $\mathcal{S}$, we write $\rho_t(s)$ the density assigned to a state $s$ after training on a sequence of states $s_1, ..., s_t$. We write $\rho'_t(s)$ the density assigned to $s$ if $\rho$ were to be trained on $s$ an additional time. The *pseudo-count* $\hat{N}$ is then defined such that updating $\rho$ on a state $s$ leads to a one unit increase of its pseudo-count

$$\rho_t(s) = \frac{\hat{N}(s)}{\hat{n}}, \quad \rho'_t(s) = \frac{\hat{N}(s) + 1}{\hat{n} + 1}, \tag{1}$$

where $\hat{n}$, the pseudo-count total is a normalization constant. This formulation of pseudo-counts match empirical counts when the density model corresponds to the empirical distribution. Equation 1 can be rewritten as

$$\hat{N}(s) = \frac{\rho_t(s)(1 - \rho'_t(s))}{\rho'_t(s) - \rho_t(s)}. \tag{2}$$

The intrinsic reward is then given by

$$r^{\text{int}}(s_t) := (\hat{N}_t(s_t))^{-1/2}. \tag{3}$$

CTS (Bellemare et al., 2014) and PixelCNN (Van den Oord et al., 2016) have been both used as density models. We will disambiguate these agent by the name of their density model.

### 2.2 INTRINSIC CURIOSITY MODULE

Intrinsic Curiosity Module (ICM, Pathak et al., 2017) promotes exploration via curiosity. Pathak et al. formulates curiosity as the error in the agent's ability to predict the consequence of its own actions in a learned feature space. ICM includes three submodules: a learned embedding, a forward and an inverse model. At the each timestep the module receives a transition $(s_t, a_t, s_{t+1})$ − where $s_t$ and $a_t$ are the current state and action and $s_{t+1}$ is the next state. States $s_t$ and $s_{t+1}$ are encoded into the features $\phi(s_t)$ and $\phi(s_{t+1})$ then passed to the inverse model which is trained to predict $a_t$. The embedding is updated at the same time, encouraging it to only model environment features that are influenced by the agent's action. The forward model has to predict the next state embedding $\phi(s_{t+1})$ using $\phi(s_t)$ and $a_t$. The intrinsic reward is then given by the error of the forward model in the embedding space between $\phi(s_{t+1})$, and the predicted estimate $\hat{\phi}(s_{t+1})$:

$$r^{\text{int}}(s_t) = \|\hat{\phi}(s_{t+1}) - \phi(s_{t+1})\|_2^2. \tag{4}$$

### 2.3 RANDOM NETWORK DISTILLATION

Random Network Distillation (RND, Burda et al., 2019) derives a bonus from the prediction error of a random network. The intuition being that the prediction error will be low on states that are similar to those previously visited and high on newly visited states. A neural network $\hat{f}$ with parameters $\theta$ is

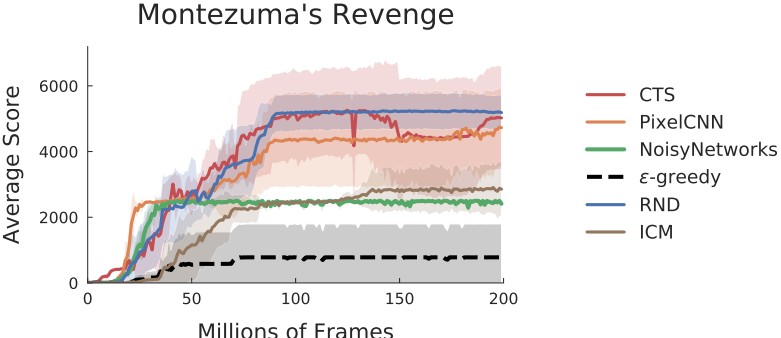

Figure 1: A comparison of different exploration methods on MONTEZUMA'S REVENGE.

trained to predict the output of a fixed, randomly initialized neural network $f$. The intrinsic reward is given by

$$r^{\text{int}}(s_t) = \|\hat{f}(s_t; \theta) - f(s_t)\|_2^2 \tag{5}$$

where $\theta$ represents the parameters of the network $\hat{f}$.

## 2.4 NOISY NETWORKS

We also evaluate Noisy Networks (NoisyNets; Fortunato et al., 2018), which is part of the original Rainbow implementation. NoisyNets does not explore using a bonus. Instead NoisyNets add noise in parameter space and replace the standard fully-connected layers, $y = Ws + b$, by a noisy version that combines a deterministic and a noisy stream:

$$y = (W + W_{noisy} \odot \epsilon^W)s + (b + b_{noisy} \odot \epsilon^b), \tag{6}$$

where $\epsilon^W$ and $\epsilon^b$ are random variables and $\odot$ denotes elementwise multiplication.

## 3 EMPIRICAL RESULTS

In this section we present our experimental study of bonus-based exploration methods.

### 3.1 EXPERIMENTAL PROTOCOL

We evaluate the three bonus-based methods introduced in Section 2 as well as NoisyNets and $\epsilon$-greedy exploration. To do so, we keep our training protocol fixed throughout our experiments. All methods are trained with a common agent architecture, the Rainbow implementation provided by the Dopamine framework (Castro et al., 2018). It includes Rainbow's three most important component: $n$-step updates (Mnih et al., 2016), prioritized experience replay (Schaul et al., 2015) and distributional reinforcement learning (Bellemare et al., 2017). Rainbow was designed combining several improvements to value-based agents that were developed independently. Pairing Rainbow with recent exploration bonuses should lead to further benefits. We also keep the original hyperparameters of the learning algorithm fixed to avoid introducing bias in favor of a specific bonus.

Unless specified, our agents are trained for 200 million frames. Following Machado et al. (2018b) recommendations we run the ALE in the stochastic setting using sticky actions ($\varsigma = 0.25$) for all agents during training and evaluation. We also do not use the mixed Monte-Carlo return (Bellemare et al., 2016; Ostrovski et al., 2017) or other algorithmic improvements that are combined with bonus-based methods (e.g. Burda et al., 2019).

### 3.2 COMMON EVALUATION FRAMEWORK

Our first set of results compare exploration methods introduced in Section 2 on MONTEZUMA'S REVENGE. We follow the standard procedure to use this game to tune the hyperparameters of each

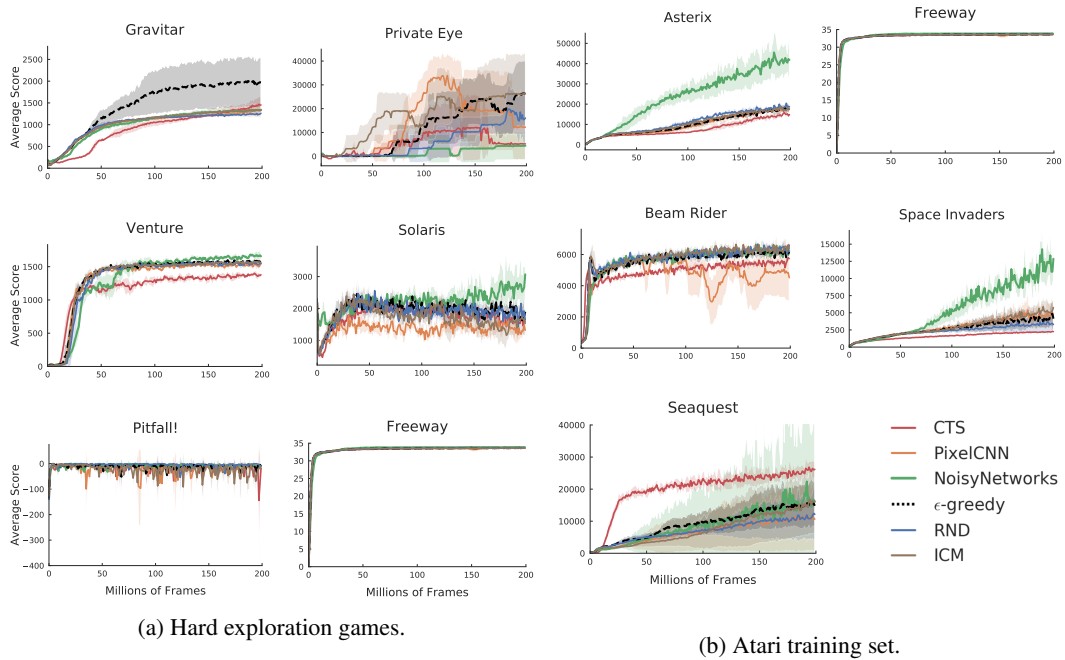

(a) Hard exploration games.

(b) Atari training set.

Figure 2: Evaluation of different bonus-based exploration methods on several Atari games, curves are averaged over 5 runs, shaded area denotes variance.

exploration bonus. Details regarding implementation and hyperparameter tuning may be found in Appendix A. Figure 1 shows training curves (averaged over 5 random seeds) for Rainbow augmented with the different exploration bonuses.

As anticipated, $\epsilon$-greedy exploration performs poorly here and struggles in such a hard exploration game. On the other hand, exploration bonuses have a huge impact and all eventually beat the baselines $\epsilon$-greedy and NoisyNets. Only ICM is unable to surpass the baselines by a large margin. RND, CTS and PixelCNN all average close to 5000 points. Interestingly, we observe that a more recent bonus like RND does not lead to higher performance over the older pseudo-count method. Of note, the performance we report at 200 millions frames improves on the performance reported in the original paper for each method. As expected, Rainbow leads to increased performance over older DQN variants previously used in the literature.

## 3.3 EXPLORATION METHODS EVALUATION

It is standard in the exploration community to, first tune hyperparameters on MONTEZUMA'S REVENGE, and then, evaluate these hyperparameters on the remaining hard exploration games with sparse rewards. This is not in line with the original ALE guidelines which recommend to evaluate on the entire suite of Atari 2600 games. In this section we provide empirical evidence that, failing to follow the ALE guidelines may interfere with the conclusions of the evaluation.

**Hard exploration games.** We now turn our attention to the set of games categorized as hard exploration games with sparse rewards in Bellemare et al.'s taxonomy (the taxonomy is available in Appendix B). It includes ALE's most difficult games for exploration, this is where good exploration strategies should shine and provide the biggest improvements. These games are: GRAVITAR, PRIVATE EYE, VENTURE, SOLARIS, PITFALL! and FREEWAY.

We evaluate agents whose hyperparameters were tuned on MONTEZUMA'S REVENGE on this set of games. Training curves averaged over 5 runs are shown in Figure 2a. We find that performance of each method on MONTEZUMA'S REVENGE does not correlate with performance on other hard exploration domains. All methods seem to behave similarly contrary to our previous observations on MONTEZUMA'S REVENGE. In particular, there is no visible difference between $\epsilon$-greedy and more sophisticated exploration bonuses. $\epsilon$-greedy exploration is on par with every other method and even

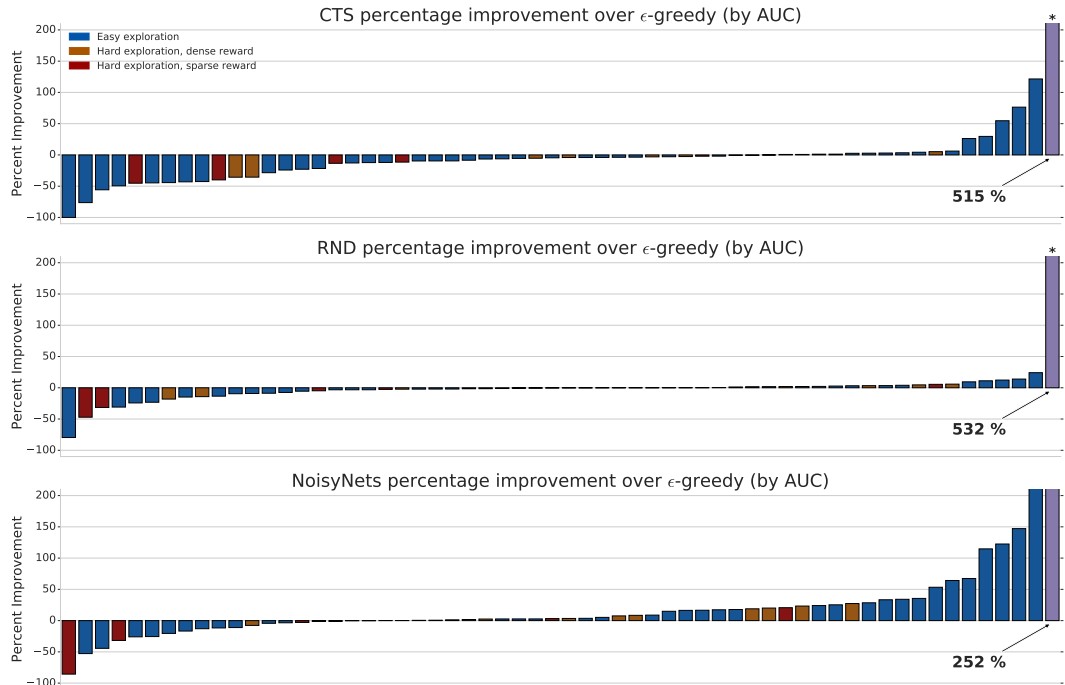

Figure 3: Improvements (in percentage of AUC) of Rainbow with various exploration methods over Rainbow with $\epsilon$-greedy exploration in 60 Atari games. The game MONTEZUMA'S REVENGE is represented in purple.

outperforms them by a significant margin on GRAVITAR. These games were initially classified to be hard exploration problems because a DQN agent using $\epsilon$-greedy exploration was unable to achieve a high scoring policy; it is no longer the case with stronger learning agents available today.

**Full Atari suite.** The set of hard exploration games was chosen to highlight the benefits of exploration methods, and, as a consequence, performance on this set may not be representative of an algorithm capabilities on the whole Atari suite. Indeed, exploration bonuses can negatively impact performance by skewing the reward landscape. To demonstrate how the choice of particular evaluation set can lead to different conclusions we also evaluate our agents on the original Atari training set which includes the games ASTERIX, FREEWAY, BEAM RIDER, SPACE INVADERS and SEAQUEST. Except for FREEWAY all these games are easy exploration problems (Bellemare et al., 2016).

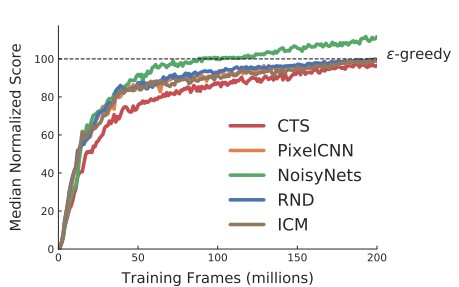

Figure 4: Normalized score of bonus-based methods compared to $\epsilon$-greedy

Figure 2b shows training curves for these games. Pseudo-counts with a CTS model appears to struggle when the environment is an easy exploration problem and end up performing worse on every game except SEAQUEST. RND and ICM are able to consistently match $\epsilon$-greedy exploration but never beat it. It appears that bonus-based methods have limited benefits in the context of easy exploration problems. Finally, despite its limited performance on MONTEZUMA'S REVENGE, we found that NoisyNets presented the most consistent improvements.

Overall, while the Atari training set and the set of the hard exploration games both show that bonus-based method only provide marginal improvements they lead us to different conclusions regarding the best performing exploration scheme. It appears that to fully quantify the behavior of exploration methods one cannot avoid evaluating on the whole Atari suite. We do so and add the remaining Atari games to our study. See Figure 3 and 4 and for a high level comparison. Altogether, it appears that

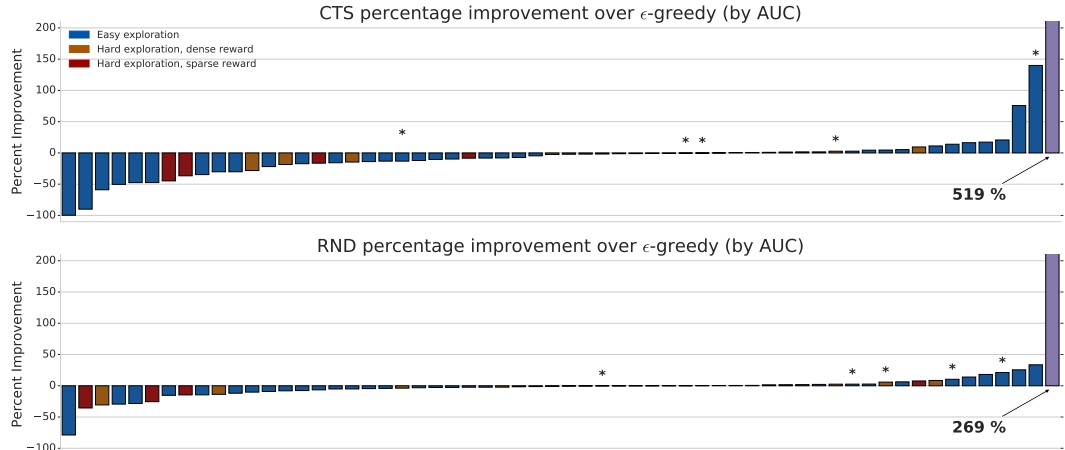

Figure 5: Improvements (in percentage of AUC) of Rainbow with CTS and RND over Rainbow with $\epsilon$-greedy exploration in 60 Atari games when hyperparameters have been tuned on SEAQUEST, QBERT, PONG, BREAKOUT and ASTERIX. The game MONTEZUMA'S REVENGE is represented in purple. Games in the training set have stars on top of their bar.

the benefit of exploration bonuses is mainly allocated towards MONTEZUMA'S REVENGE. None of them, except NoisyNets, seems to improve over $\epsilon$-greedy by significant margin. Bonus-based methods may lead to increased performance on a few games but seem to deteriorate performance by a roughly equal amount on other games.

We could have hoped that these methods focus their attention on hard exploration games at the expense of easier ones, meaning they trade exploitation for exploration. Unfortunately, it does not seem to be the case as they do not exhibit a preference for hard exploration games (red and orange bars). We may wonder if these methods are overfitting on MONTEZUMA'S REVENGE.

### 3.4 Hyperparameter tuning procedure

Previous experiments have so far depicted a somewhat dismal picture of bonus-based methods, in particular their penchant to overfit on MONTEZUMA'S REVENGE. Though it is unfortunate to see they do not generalize to the full Atari gamut, one may wonder if their tendency to overfit to MONTEZUMA'S REVENGE specifically is caused by the hyperparameter tuning procedure or their inherent design. The experiments in this section aim at addressing this issue. We ran a new hyperparameter sweep on CTS and RND using a new training set. We chose these algorithms as we found them to be particularly sensitive to their hyperparameters. The new training set includes the games PONG, ASTERIX, SEAQUEST, Q*BERT and BREAKOUT. This set as been previously used by others as a training set to tune reinforcement learning agents (Bellemare et al., 2017; Dabney et al., 2018).

Results are depicted in Figure 5. Both algorithm still perform much better than $\epsilon$-greedy exploration on MONTEZUMA'S REVENGE. As it is to be expected, performance also increased for games within the training set. A notable example is CTS on SEAQUEST which now improves over the $\epsilon$-greedy baseline by 140% instead of 120% previously. Nevertheless, the conclusions from Section 3.3 remain valid. Bonus-based exploration strategies still provide only limited value except for a few games. In addition, neither algorithm seems to achieve better results on easy exploration problems outside of the training set.

### 3.5 Amount of training data

Current results so far showed that exploration bonuses provide little benefits in the context of exploration in the ALE. Nonetheless, our experiments have been limited to 200 millions environment frames, it is possible that with additional data exploration bonuses would perform better. This would be in line with an emerging trend of training agents an order of magnitude longer in order to produce

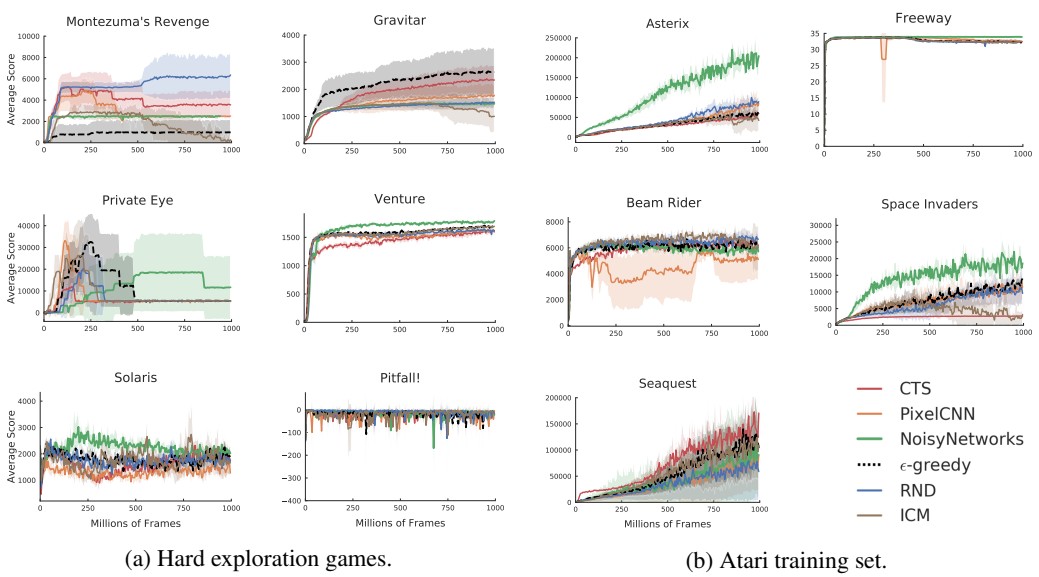

(a) Hard exploration games.      (b) Atari training set.

Figure 6: Evaluation of bonus-based methods when training is extended to one billion frames

a high-scoring policy, irrespective of the sample cost (Espeholt et al., 2018; Burda et al., 2019; Kapturowski et al., 2019). In this section we present results that contradict this hypothesis. We reuse agents trained in Section 4.2.2 and lengthen the amount training data they process to 1 billion environment frames. We use Atari 2600 games from the set hard exploration with sparse rewards and the Atari training set. See Figure 6 for training curves. All exploration strategies see their score gracefully scale with additional data on easier exploration problems. In hard exploration games, none of the exploration strategies seem to benefit from receiving more data. Score on most games seem to plateau and may even decrease. This is particularly apparent in MONTEZUMA'S REVENGE where only RND actually benefits from longer training. After a while, agents are unable to make further progress and their bonus may collapse. When that happens, bonus-based methods cannot even rely on $\epsilon$-greedy exploration to explore and may therefore see their performance decrease. This behavior may be attributed to our evaluation setting, tuning hyperparameters to perform best after one billion training frames will likely improve results. It is however unsatisfactory to see that exploration methods do not scale effortlessly as they receive more data. In practice, recent exploration bonuses have required a particular attention to handle large amount of training data (e.g. Burda et al., 2019). Moreover, it is interesting to see that the performance at 200 millions frames leads to conclusions similar to those obtained with one billion frames.

## 4 CONCLUSION

Recently, many exploration methods have been introduced with confounding factors within their evaluation – longer training duration, different model architecture and other ad-hoc techniques. This practice obscures the true value of these methods and makes it difficult to prioritize more promising directions of investigation. Therefore, following a growing trend in the reinforcement learning community, we advocate for better practices on empirical evaluation for exploration to fairly assess the contribution of newly proposed methods. In a standardized training environment and context, we found that current bonus-based methods are unable to surpass $\epsilon$-greedy exploration. As a whole this suggest progress in bonus-based exploration may have been driven by confounding factors rather than improvement in the bonuses. This shows that more work is still needed to address the exploration problem in complex environments.

We may wonder why do we not see a positive impact of bonus-based exploration. One possible explanation is our use of the relatively sophisticated Rainbow learning algorithm. The benefits that others have observed using exploration bonuses might be made redundant through other mechanisms already in Rainbow, such as a prioritized replay buffer. An important transition may only be observed

with low frequency but it can be sampled at a much higher rate by the replay buffer. As a result, the agent can learn from it effectively without the need for encouragement from the exploration bonus to visit that transition more frequently. Though the merits of bonus-based methods have been displayed with less efficient learning agents, these benefits did not carry on to improved learning agents. This is disappointing given that the simple NoisyNets baseline showed efficient exploration can still achieve noticeable gains on the ALE. As of now, the exploration and credit assignment communities have mostly been operating independently. Our work suggests that they may have to start working hand in hand to design new agents to make reinforcement learning truly sample efficient.

ACKNOWLEDGEMENTS

The authors would like to thank Hugo Larochelle, Benjamin Eysenbach, Danijar Hafner, Valentin Thomas and Ahmed Touati for insightful discussions as well as Sylvain Gelly for careful reading and comments on an earlier draft of this paper. This work was supported by the CIFAR Canadian AI Chair.

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

## A  Hyperparameter tuning

All bonus based methods are tuned with respect to their final performance on Montezuma's Revenge after training on 200 million frames averaged over five runs. When different hyper parameter settings led to comparable final performance we chose the one that achieve the performance the fastest.

### A.1  Rainbow and Atari preprocessing

We used the standard Atari preprocessing from Mnih et al. (2015). Following Machado et al. (2018b) recommendations we enable sticky actions and deactivated the termination on life loss heuristic. The remaining hyperparameters were chosen to match Hessel et al. (2018) implementation.

| Hyperparameter | Value |
|---|---|
| Discount factor $\gamma$ | 0.99 |
| Min history to start learning | 80K frames |
| Target network update period | 32K frames |
| Adam learning rate | $6.25 \times 10^{-5}$ |
| Adam $\epsilon$ | $1.5 \times 10^{-4}$ |
| Multi-step returns $n$ | 3 |
| Distributional atoms | 51 |
| Distributional min/max values | [-10, 10] |

Every method except NoisyNets is trained with $\epsilon$-greedy following the scheduled used in Rainbow with $\epsilon$ decaying from 1 to 0.01 over 1M frames.

### A.2  Hyperparameter tuning on Montezuma's Revenge

#### A.2.1  NoisyNets

We did not tune NoisyNets. We kept the original hyperparameter $\sigma_0 = 0.5$ as in Fortunato et al. (2018) and Hessel et al. (2018).

#### A.2.2  Pseudo-counts

We followed Bellemare et al.'s preprocessing, inputs are $42 \times 42$ greyscale images, with pixel values quantized to 8 bins.

**CTS**: We tuned the scaling factor $\beta$. We ran a sweep for $\beta \in \{0.5, 0.1, 0.05, 0.01, 0.005, 0.001, 0.0005, 0.0001\}$ and found that $\beta = 0.0005$ worked best.

**PixelCNN**: We tuned the scaling factor and the prediction gain decay constant $c$. We ran a sweep with the following values: $\beta \in \{5.0, 1.0, 0.5, 0.1, 0.05\}$, $c \in \{5.0, 1.0, 0.5, 0.1, 0.05\}$ and found $\beta = 0.1$ and $c = 1.0$ to work best.

### A.3  ICM

We tuned the scaling factor and the scalar $\alpha$ that weighs the inverse model loss against the forward model. We ran a sweep with $\alpha = \{0.4, 0.2, 0.1, 0.05, 0.01, 0.005\}$ and $\beta = \{2.0, 1.0, 0.5, 0.1, 0.05, 0.01, 0.005, 0.001, 0.0005\}$. We found $\alpha = 0.005$ and $\beta = 0.005$ to work best.

### A.4  RND

Following Burda et al. (2019) we did not clip the intrinsic reward while the extrinsic reward was clipped. We tuned the reward scaling factor $\beta$ and learning rate used in Adam (Kingma & Ba, 2014) by the RND optimizer. We ran a sweep with $\beta = \{0.05, 0.01, 0.005, 0.001, 0.0005, 0.0001, 0.0005, 0.00001, 0.000005\}$ and lr $= \{0.005, 0.001, 0.0005, 0.0002, 0.0001, 0.00005\}$. We found that $\beta = 0.00005$ and lr $= 0.0001$ worked best.

## A.5 HYPERPARAMETER TUNING ON SECOND TRAINING SET (SECTION 3.4)

### A.5.1 PSEUDO-COUNTS CTS

We ran a sweep for $\beta \in \{0.1, 0.05, 0.01, 0.005, 0.001, 0.0005, 0.0001\}$ and found that $\beta = 0.0001$ worked best.

## A.6 RND

We did not tune the learning rate and kept lr $=$ 0.0001. We ran a sweep for $\beta =$ $\{[0.05, 0.01, 0.005, 0.001, 0.0005, 0.0001, 0.00005\}$ and found that $\beta = 0.00005$ worked best.

## B TAXONOMY OF EXPLORATION ON THE ALE

Bellemare et al.'s taxonomy of exploration propose a classification of Atari 2600 games in terms of difficulty of exploration. It is provided in Table 1.

Table 1: A classification of Atari 2600 games based on their exploration difficulty.

| Easy Exploration | | | Hard exploration | |
|---|---|---|---|---|
| Human Optimal | | Score Exploit | Dense Reward | Sparse Reward |
| ASSAULT | ASTERIX | BEAM RIDER | ALIEN | FREEWAY |
| ASTEROIDS | ATLANTIS | KANGAROO | AMIDAR | GRAVITAR |
| BATTLE ZONE | BERZERK | KRULL | BANK HEIST | MONTEZUMA'S REVENGE |
| BOWLING | BOXING | KUNG-FU MASTER | FROSTBITE | PITFALL! |
| BREAKOUT | CENTIPEDE | ROAD RUNNER | H.E.R.O | PRIVATE EYE |
| CHOPPER CMD | CRAZY CLIMBER | SEAQUEST | MS. PAC-MAN | SOLARIS |
| DEFENDER | DEMON ATTACK | UP N DOWN | Q*BERT | VENTURE |
| DOUBLE DUNK | ENDURO | TUTANKHAM | SURROUND | |
| FISHING DERBY | GOPHER | | WIZARD OF WOR | |
| ICE HOCKEY | JAMES BOND | | ZAXXON | |
| NAME THIS GAME | PHOENIX | | | |
| PONG | RIVER RAID | | | |
| ROBOTANK | SKIING | | | |
| SPACE INVADERS | STARGUNNER | | | |

## C ADDITIONAL FIGURES

The variance of the return on MONTEZUMA'S REVENGE is high because the reward is a step function, for clarity we also provide all the training curves in Figure 7

We also provide training curves presented in the main paper in larger format.

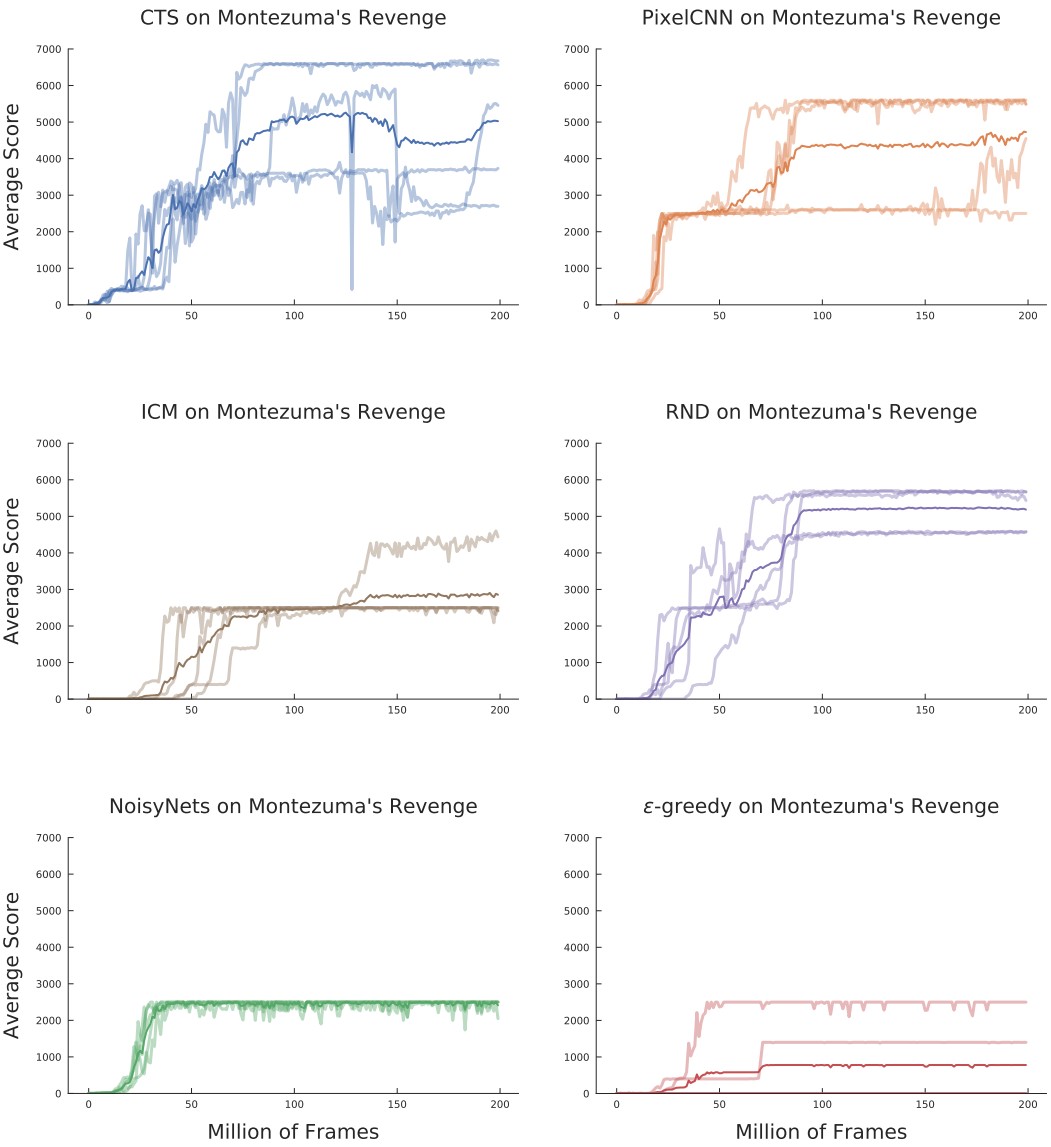

Figure 7: Training curves on MONTEZUMA'S REVENGE

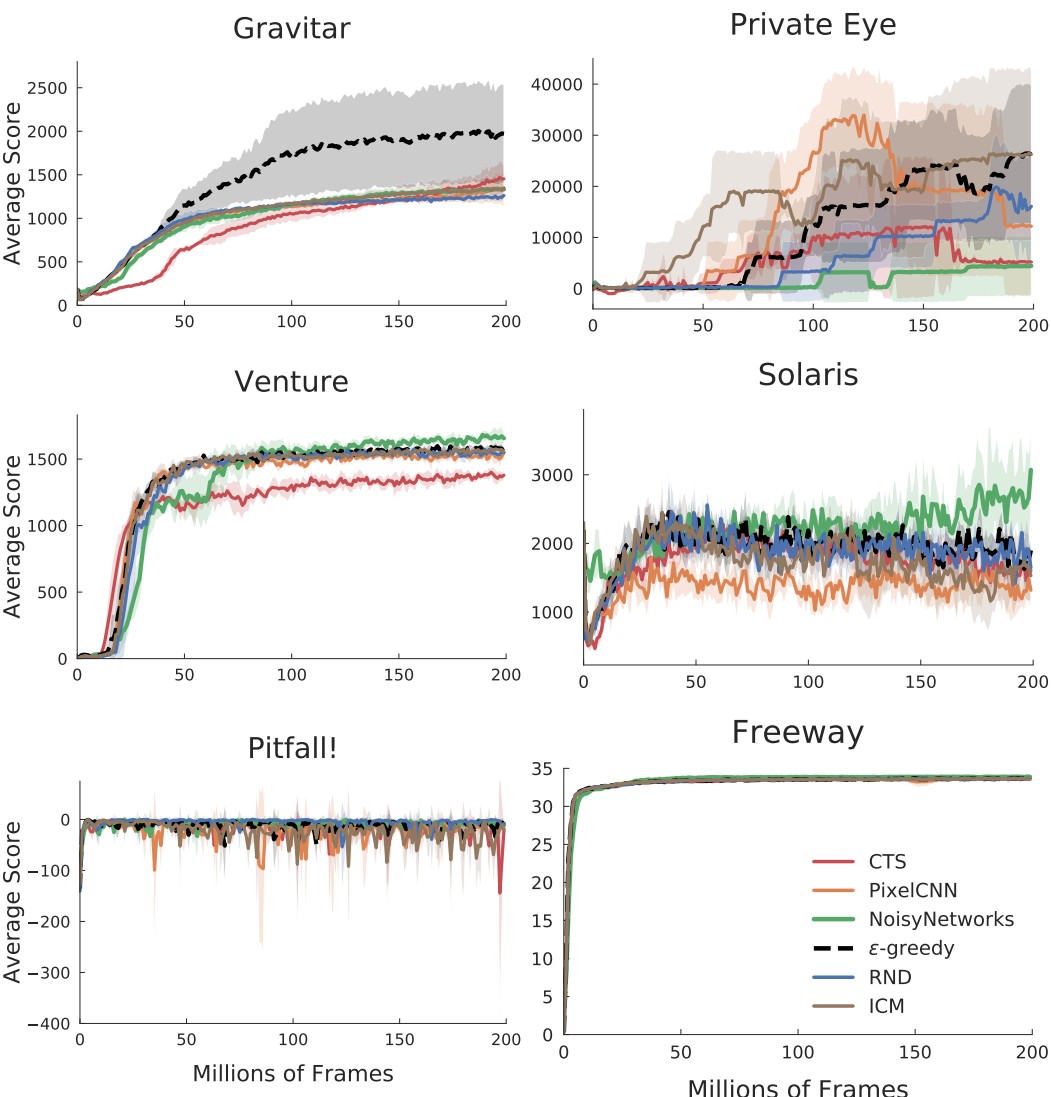

Figure 8: Evaluation of different bonus-based exploration methods on the set of hard exploration games with sparse rewards. Curves are average over 5 runs and shaded area denotes variance.

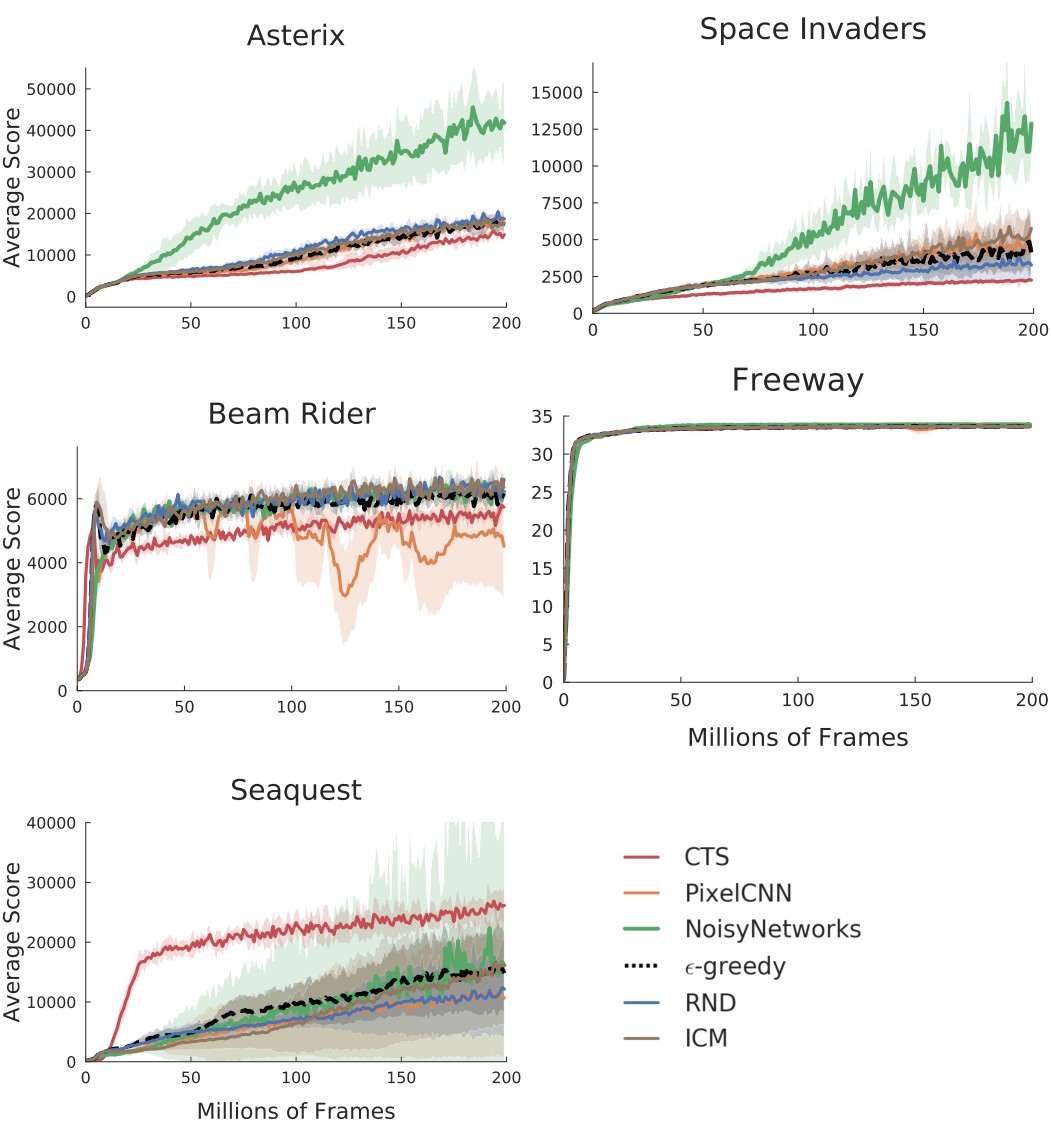

Figure 9: Evaluation of different bonus-based exploration methods on the Atari training set. Curves are average over 5 runs and shaded area denotes variance.

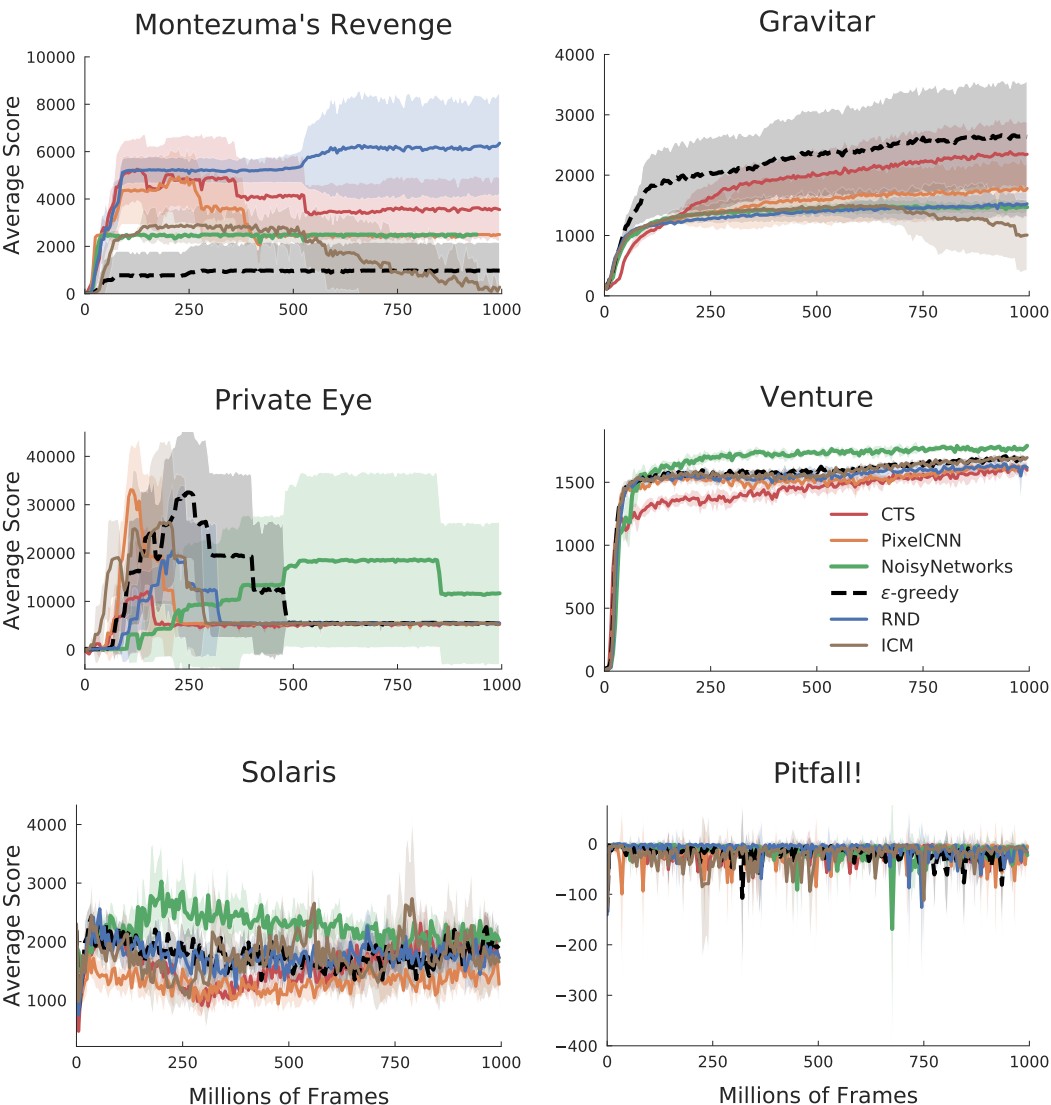

Figure 10: Evaluation of different bonus-based exploration methods on the set of hard exploration games with sparse rewards. Exploration methods are trained for one billion frames. Curves are average over 5 runs and shaded area represents variance.

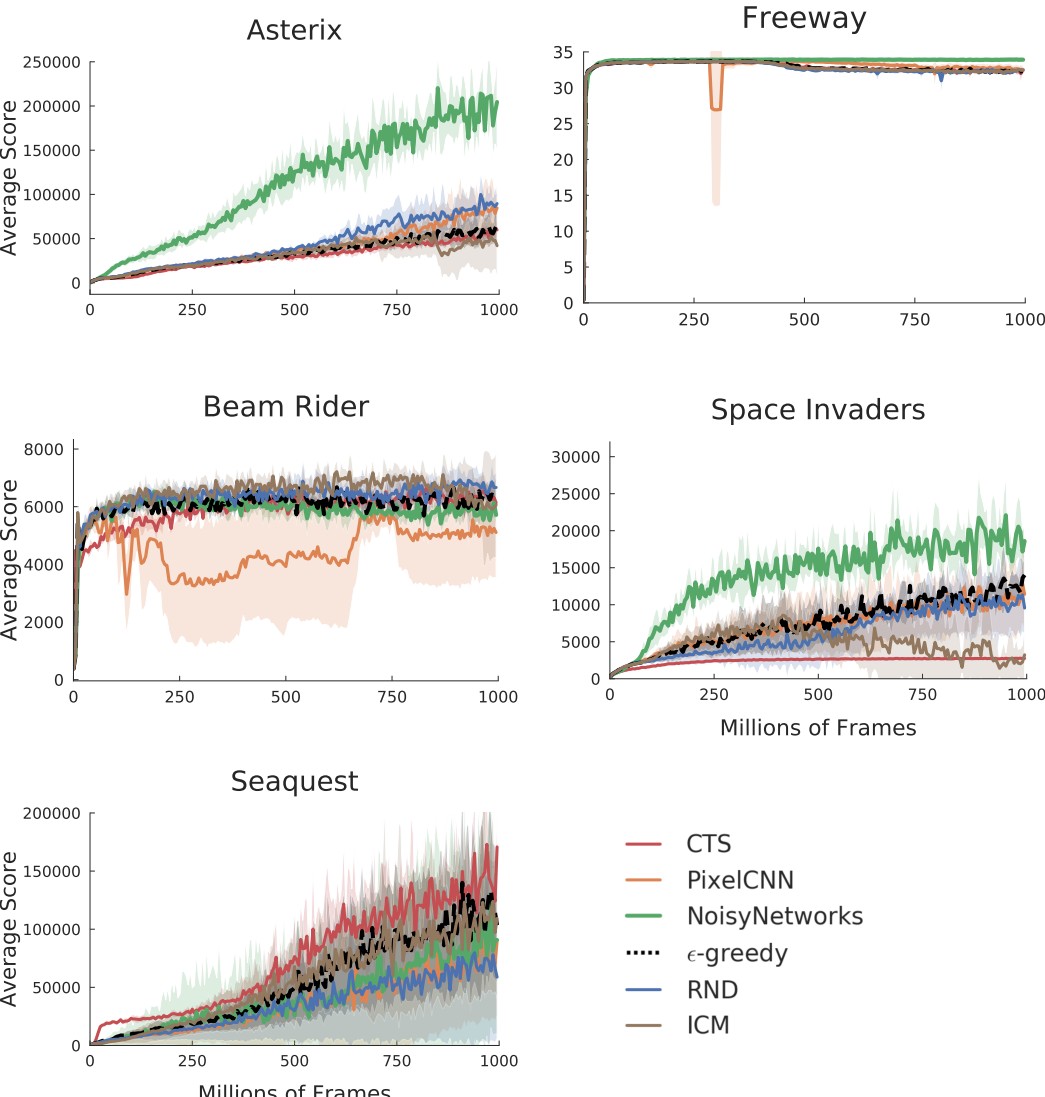

Figure 11: Evaluation of different bonus-based exploration methods on the Atari training set. Exploration methods are trained for one billion frames. Curves are average over 5 runs and shaded area represents variance.

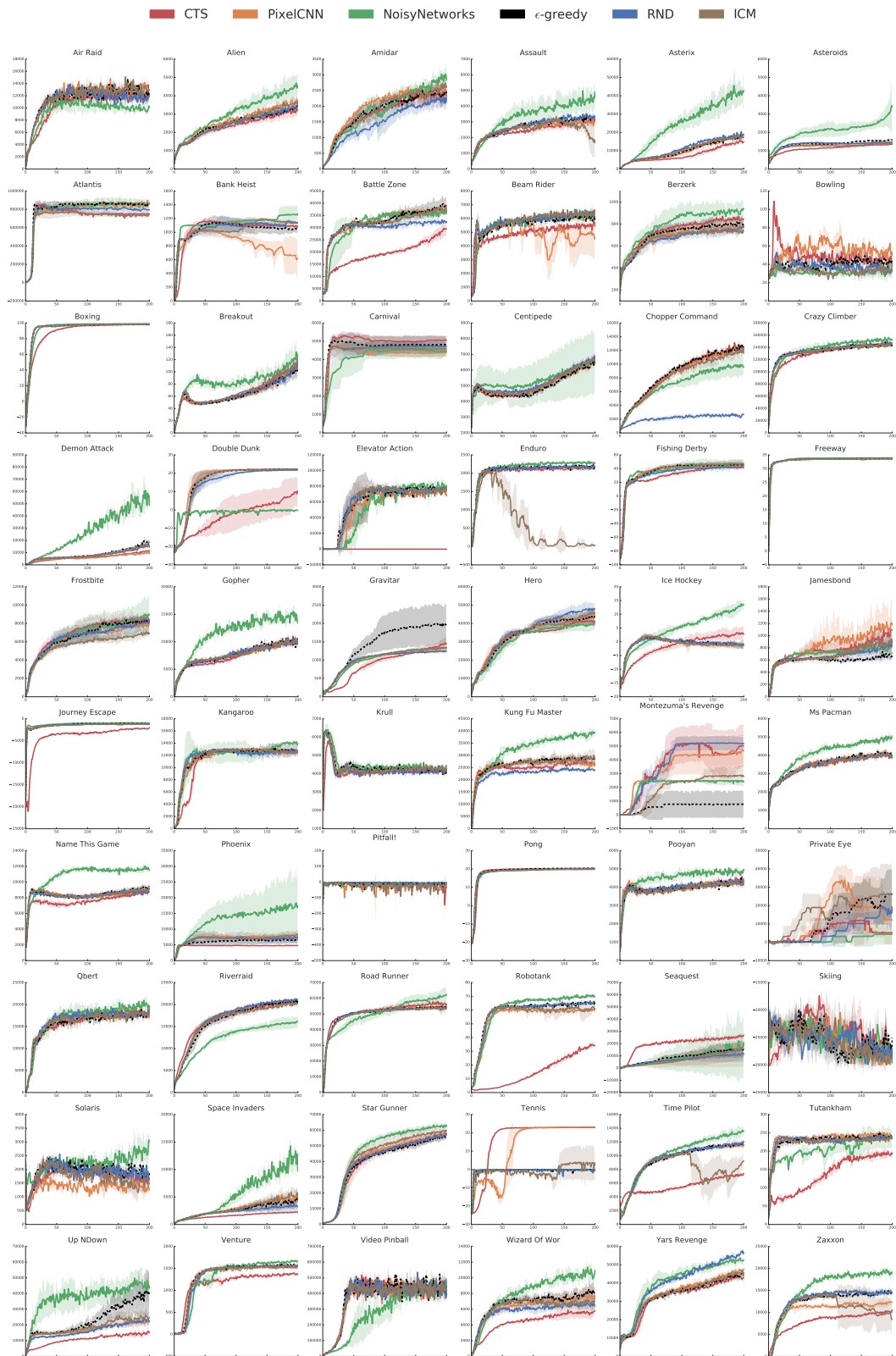

Figure 12: Training curves of Rainbow with $\epsilon$-greedy, CTS, PixelCNN, NoisyNets, ICM and RND.

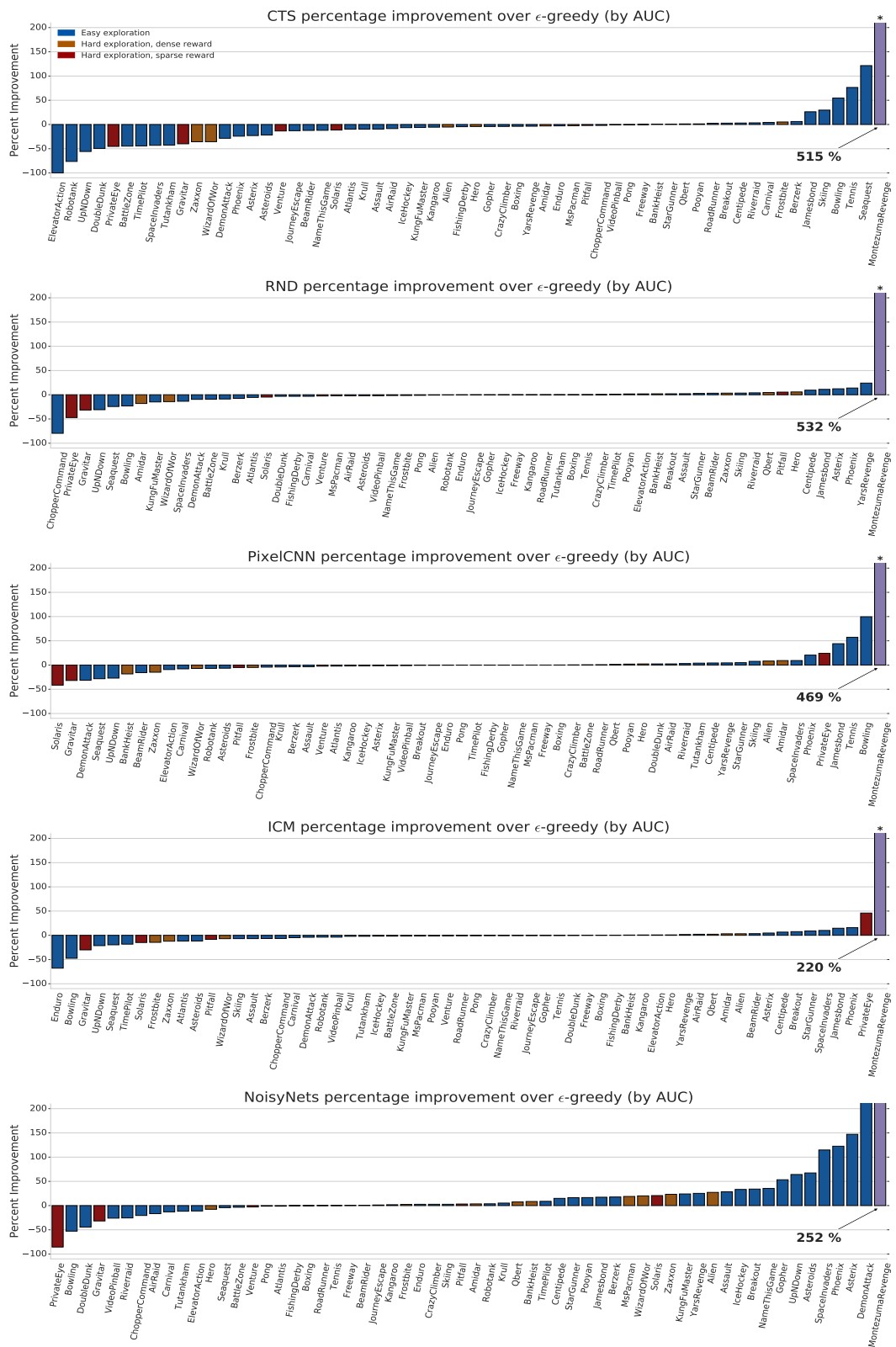

Figure 13: Improvements (in percentage of AUC) of Rainbow with various exploration methods over Rainbow with ε-greedy exploration in 60 Atari games when hyperparameters are tuned on MONTEZUMA'S REVENGE. The game MONTEZUMA'S REVENGE is represented in purple.

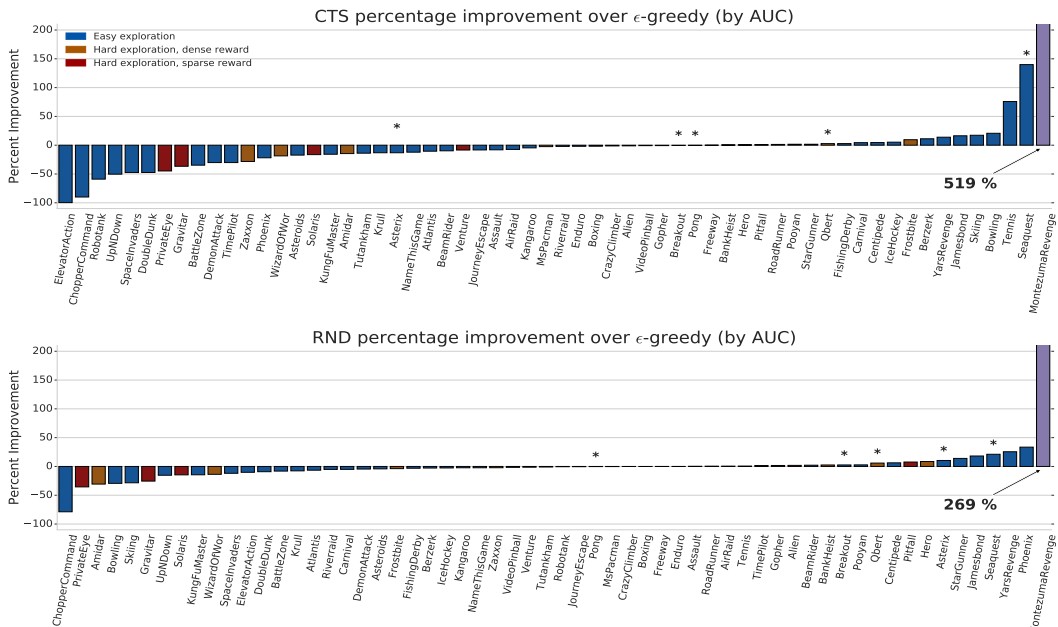

Figure 14: Improvements (in percentage of AUC) of Rainbow with CTS and RND over Rainbow with ε-greedy exploration in 60 Atari games when hyperparameters have been tuned on SEAQUEST, QBERT, PONG, BREAKOUT and ASTERIX. The game MONTEZUMA'S REVENGE is represented in purple. Games in the training set have stars on top of their bar.

Table 2: Raw scores for all games after 10M training frames

| Games | $\epsilon$-greedy | RND | CTS | PixelCNN | ICM | NoisyNets |
|---|---|---|---|---|---|---|
| AirRaid | 5393.2 | 5374.7 | 4084.7 | **5425.8** | 5374.7 | 4361.4 |
| Alien | 1289.6 | **1352.1** | 1250.6 | 1228.0 | 1310.6 | 1222.0 |
| Amidar | 335.0 | 286.8 | 345.1 | **428.0** | 364.9 | 324.7 |
| Assault | 1459.5 | 1453.4 | 1303.6 | **1641.9** | 1431.7 | 1080.4 |
| Asterix | 2777.5 | 2793.9 | **2841.0** | 2748.0 | 2736.7 | 2698.6 |
| Asteroids | 759.6 | 848.4 | 722.7 | 800.5 | 789.5 | **1117.0** |
| Atlantis | 117166.5 | 172134.9 | 99510.6 | **184420.5** | 148927.5 | 109464.9 |
| BankHeist | 901.5 | 896.5 | 426.7 | 756.8 | 904.4 | **1039.9** |
| BattleZone | **23909.5** | 22374.9 | 9734.0 | 22822.1 | 22575.3 | 11581.6 |
| BeamRider | 5475.7 | 5135.5 | 4066.8 | 3415.4 | **5796.6** | 3845.6 |
| Berzerk | 419.1 | 442.5 | 446.8 | 435.5 | 430.8 | **476.0** |
| Bowling | 40.8 | 39.0 | **92.6** | 47.6 | 35.5 | 27.1 |
| Boxing | 75.7 | 74.8 | 45.2 | **76.0** | 74.7 | 57.5 |
| Breakout | 49.2 | **49.5** | 45.6 | 46.4 | 48.8 | 44.7 |
| Carnival | **4644.9** | 4375.3 | 3658.4 | 4138.8 | 4317.0 | 2291.5 |
| Centipede | 4845.3 | 5009.4 | 4731.0 | 4963.4 | **5110.2** | 4912.0 |
| ChopperCommand | **2494.5** | 913.6 | 2293.9 | 2420.5 | 2376.6 | 1899.1 |
| CrazyClimber | 107706.9 | 105940.9 | 90549.0 | **107724.2** | 104783.2 | 96063.8 |
| DemonAttack | 2932.2 | 3135.8 | 1442.0 | 2564.6 | **3153.3** | 1864.6 |
| DoubleDunk | -18.4 | -18.1 | -19.4 | -18.2 | -19.0 | **-8.2** |
| ElevatorAction | 29.1 | 114.7 | 0.0 | **141.8** | 22.9 | 0.0 |
| Enduro | 1657.1 | 1719.1 | 1230.0 | **1767.0** | 1727.4 | 1272.9 |
| FishingDerby | **15.5** | 9.7 | 13.8 | 15.0 | 14.5 | 10.8 |
| Freeway | **32.3** | 32.2 | 32.2 | 32.3 | 32.2 | 31.7 |
| Frostbite | 3035.7 | **3286.5** | 3078.9 | 2787.7 | 2854.7 | 2913.0 |
| Gopher | 4047.6 | 4224.1 | 4295.8 | **4349.7** | 4266.1 | 3760.1 |
| Gravitar | 229.4 | **236.8** | 130.0 | 235.1 | 225.8 | 211.3 |
| Hero | 10730.5 | 10858.1 | 12753.7 | **12999.9** | 11808.4 | 9441.1 |
| IceHockey | -5.0 | **-4.9** | -11.9 | -5.9 | -5.1 | -7.7 |
| Jamesbond | 520.3 | 508.0 | 473.2 | **536.0** | 517.0 | 484.6 |
| JourneyEscape | -2376.4 | -2299.0 | -9208.8 | -2463.4 | -2414.1 | **-1784.3** |
| Kangaroo | 5233.1 | 5537.0 | 1747.0 | 5266.2 | **5823.1** | 5147.0 |
| Krull | 5941.1 | 5997.8 | 5620.4 | 5999.8 | 5935.5 | **6195.1** |
| KungFuMaster | 19824.1 | 17676.9 | **22668.2** | 21673.9 | 19199.3 | 19150.5 |
| MontezumaRevenge | 0.5 | 1.5 | **231.8** | 17.8 | 0.0 | 13.2 |
| MsPacman | 2371.0 | 2271.4 | **2452.5** | 2443.6 | 2424.6 | 2436.4 |
| NameThisGame | 9086.7 | 8561.7 | 7402.8 | **9089.4** | 8820.6 | 7590.1 |
| Phoenix | 4758.3 | 4790.7 | 4498.7 | 4840.8 | 4788.1 | **4868.0** |
| Pitfall | -13.3 | -10.4 | **-2.4** | -11.1 | -5.6 | -3.0 |
| Pong | 14.6 | 14.0 | 9.8 | 14.3 | **15.5** | 12.4 |
| Pooyan | 3917.3 | 3867.7 | 3885.9 | **3996.4** | 3722.1 | 3623.4 |
| PrivateEye | 77.4 | **180.3** | -903.9 | 89.1 | 116.5 | 78.9 |
| Qbert | 7025.6 | 7442.4 | 5466.8 | 5885.5 | **8090.2** | 4363.0 |
| Riverraid | 4578.2 | 4904.3 | **6830.2** | 5002.2 | 4739.3 | 3820.5 |
| RoadRunner | 33743.4 | 34944.0 | **39145.9** | 35081.6 | 34109.7 | 33649.4 |
| Robotank | 22.5 | **22.7** | 1.8 | 17.9 | 22.6 | 18.5 |
| Seaquest | 1711.6 | **1744.0** | 1476.0 | 1631.9 | 1647.4 | 1155.9 |
| Skiing | -22580.5 | **-21378.9** | -27888.8 | -23652.6 | -23040.9 | -22729.0 |
| Solaris | 1164.7 | 1103.4 | 1108.2 | 966.8 | 1254.2 | **1495.9** |
| SpaceInvaders | 793.6 | 764.5 | 641.6 | **800.6** | 780.6 | 687.0 |
| StarGunner | 1533.2 | 1523.3 | 1579.6 | 1530.3 | 1500.3 | **1587.5** |
| Tennis | -2.8 | **-0.4** | -21.2 | -7.5 | -1.0 | -1.0 |
| TimePilot | 2969.7 | 3001.6 | **4516.1** | 2810.5 | 3049.9 | 2710.6 |
| Tutankham | 187.0 | 207.2 | 54.3 | **232.3** | 169.2 | 162.2 |
| UpNDown | 14237.9 | 11060.1 | 4373.8 | 12494.2 | 13220.7 | **16120.4** |
| Venture | 25.0 | 16.4 | 13.3 | 17.8 | 19.0 | **34.9** |
| VideoPinball | **33347.6** | 30284.0 | 31173.0 | 33294.5 | 30661.6 | 22802.8 |
| WizardOfWor | 2432.5 | **2865.3** | 1451.5 | 2376.6 | 2191.3 | 1568.7 |
| YarsRevenge | 10349.2 | 11161.2 | 9926.9 | 11002.0 | **11415.6** | 10930.2 |
| Zaxxon | 4282.7 | **4707.7** | 3603.9 | 3766.4 | 4266.8 | 3016.6 |

Table 3: Raw scores for all games after 50M training frames

| Games | $\epsilon$-greedy | RND | CTS | PixelCNN | ICM | NoisyNets |
|---|---|---|---|---|---|---|
| AirRaid | 12081.8 | **13005.2** | 11016.9 | 12399.0 | 11392.8 | 10433.7 |
| Alien | 2114.1 | 2004.9 | 1969.4 | 2200.6 | 1979.4 | **2308.3** |
| Amidar | 1535.6 | 1140.6 | 1469.5 | **1791.9** | 1504.0 | 1337.2 |
| Assault | 2384.1 | 2423.8 | 2265.6 | 2465.2 | 2555.6 | **3050.1** |
| Asterix | 5722.4 | 6488.7 | 4959.4 | 5439.3 | 5931.7 | **13981.5** |
| Asteroids | 1363.7 | 1409.2 | 1095.8 | 1315.3 | 1247.5 | **1901.6** |
| Atlantis | **831286.0** | 807958.0 | 805574.0 | 816578.0 | 763538.0 | 827702.0 |
| BankHeist | 1113.8 | 1070.0 | **1154.6** | 1005.5 | 1028.9 | 1104.3 |
| BattleZone | 30959.7 | 31271.5 | 16167.4 | 31011.8 | **32357.5** | 30606.6 |
| BeamRider | 5550.9 | **5782.2** | 4726.8 | 5504.9 | 5633.0 | 5636.4 |
| Berzerk | 697.9 | 643.8 | 700.4 | 662.8 | 667.5 | **731.7** |
| Bowling | 41.4 | 42.6 | 45.5 | **56.5** | 36.1 | 30.0 |
| Boxing | 95.7 | 95.8 | 91.2 | 95.8 | 96.0 | **98.0** |
| Breakout | 48.3 | 48.3 | 48.6 | 49.7 | 51.2 | **82.3** |
| Carnival | 4936.0 | 4554.8 | **5199.7** | 4600.6 | 4644.4 | 4110.6 |
| Centipede | 4312.1 | 4568.4 | 4235.0 | 4544.2 | 4652.0 | **4986.9** |
| ChopperCommand | 6438.2 | 2087.1 | **6504.8** | 6010.3 | 5840.4 | 5589.4 |
| CrazyClimber | 131002.1 | 132982.7 | 126655.8 | 133039.8 | 130491.3 | **133873.7** |
| DemonAttack | 6021.3 | 5590.7 | 4499.4 | 4985.7 | 6052.2 | **13754.6** |
| DoubleDunk | 17.9 | 15.4 | -11.0 | **20.0** | 18.0 | -0.6 |
| ElevatorAction | 55982.0 | **66996.0** | 0.0 | 52254.0 | 58349.4 | 17514.5 |
| Enduro | 2173.9 | 2076.4 | 2105.2 | 2119.4 | 1683.3 | **2180.9** |
| FishingDerby | 35.1 | 32.7 | 23.9 | 37.2 | **37.5** | 33.6 |
| Freeway | 33.4 | 33.3 | 33.3 | 33.3 | 33.4 | **33.7** |
| Frostbite | 5773.9 | 6162.9 | **6334.9** | 5759.3 | 5551.6 | 6010.9 |
| Gopher | 6564.7 | 6445.1 | 6111.8 | 6688.1 | 6761.3 | **10255.0** |
| Gravitar | **1148.3** | 990.6 | 635.6 | 953.9 | 950.7 | 903.5 |
| Hero | 29014.5 | 30679.0 | 29273.5 | **31150.1** | 30756.5 | 27165.2 |
| IceHockey | 1.3 | 1.0 | -3.8 | 1.8 | **1.8** | 0.9 |
| Jamesbond | 600.8 | 633.0 | 669.5 | **712.6** | 622.9 | 707.7 |
| JourneyEscape | -1448.3 | -1422.1 | -3424.2 | -1437.9 | -1541.1 | **-1304.6** |
| Kangaroo | 12426.5 | **12731.0** | 12333.7 | 11771.7 | 12606.5 | 12428.5 |
| Krull | **4522.9** | 4065.4 | 4041.7 | 4051.8 | 4319.1 | 4341.5 |
| KungFuMaster | 26929.2 | 22972.3 | 25573.2 | 26135.5 | 25883.8 | **30161.7** |
| MontezumaRevenge | 553.3 | **2789.6** | 2550.4 | 2528.5 | 1160.7 | 2492.9 |
| MsPacman | 3185.3 | 3218.8 | 3272.4 | 3400.7 | 3310.0 | **3768.5** |
| NameThisGame | 8304.9 | 8106.9 | 6978.1 | 8467.3 | 8222.7 | **10883.2** |
| Phoenix | 5862.1 | 7434.0 | 4781.2 | 7664.7 | 7369.7 | **11873.4** |
| Pitfall | -4.0 | **-3.6** | -13.9 | -10.1 | -13.1 | -4.5 |
| Pong | 19.5 | 19.0 | **19.9** | 19.2 | 19.1 | 19.2 |
| Pooyan | 3703.3 | 3884.6 | 3724.3 | 3727.1 | 3775.3 | **4561.7** |
| PrivateEye | 127.6 | 319.6 | 948.6 | 679.7 | **12156.0** | 96.2 |
| Qbert | 15498.0 | **17014.5** | 14804.3 | 16463.1 | 16786.7 | 16780.0 |
| Riverraid | 14935.6 | 15797.5 | 15747.2 | **16137.5** | 15590.9 | 10469.4 |
| RoadRunner | 50559.5 | 50349.6 | **50915.7** | 49905.1 | 50042.5 | 46314.5 |
| Robotank | 62.2 | 62.0 | 4.6 | 59.4 | 60.5 | **63.9** |
| Seaquest | 4915.9 | 5036.7 | **19227.6** | 3708.0 | 3331.5 | 4433.2 |
| Skiing | **-21899.7** | -24633.1 | -23028.8 | -25674.1 | -24216.7 | -25306.5 |
| Solaris | 2272.7 | **2347.2** | 1921.1 | 1444.2 | 2076.0 | 2063.7 |
| SpaceInvaders | 1904.4 | 1881.2 | 1275.9 | **2007.6** | 1827.3 | 1888.8 |
| StarGunner | 35867.0 | 38744.5 | 33868.7 | 39416.2 | 41771.6 | **42542.3** |
| Tennis | -0.1 | 0.0 | **22.1** | -14.3 | -0.7 | -1.2 |
| TimePilot | 8901.6 | **9295.9** | 4785.5 | 8927.9 | 9220.9 | 8956.7 |
| Tutankham | 227.9 | **235.1** | 96.6 | 234.6 | 220.8 | 201.8 |
| UpNDown | 15097.4 | 13116.4 | 9151.3 | 14436.6 | 14621.5 | **39546.4** |
| Venture | **1460.4** | 1435.4 | 1117.7 | 1426.2 | 1443.2 | 1220.3 |
| VideoPinball | 398511.1 | 419047.9 | 414727.1 | 386829.1 | **452874.2** | 154968.4 |
| WizardOfWor | 6941.0 | 5932.6 | 4108.2 | 6226.2 | **7027.8** | 6906.1 |
| YarsRevenge | 30769.2 | 36086.1 | 30596.5 | 32049.2 | 30721.4 | **36136.7** |
| Zaxxon | 13161.2 | 13341.5 | 7223.7 | 11332.8 | 13177.1 | **13936.2** |

Table 4: Raw scores for all games after 100M training frames

| Games | $\epsilon$-greedy | RND | CTS | PixelCNN | ICM | NoisyNets |
|---|---|---|---|---|---|---|
| AirRaid | 11744.3 | 12081.2 | 11819.6 | **13292.9** | 12744.2 | 10096.6 |
| Alien | 2696.7 | 2531.9 | 2519.4 | 2761.6 | 2593.9 | **3237.0** |
| Amidar | 2033.1 | 1527.9 | 1797.0 | **2295.1** | 2167.2 | 1872.9 |
| Assault | 2946.8 | 2832.5 | 2535.3 | 2839.1 | 2574.8 | **3718.9** |
| Asterix | 9279.4 | 10785.7 | 6083.1 | 8540.0 | 10661.5 | **26367.6** |
| Asteroids | 1374.3 | 1423.3 | 1156.8 | 1391.8 | 1266.7 | **2169.9** |
| Atlantis | **859040.0** | 799144.2 | 747436.0 | 852382.0 | 754256.0 | 857076.0 |
| BankHeist | 1106.0 | 1090.6 | 1131.7 | 932.2 | 1013.4 | **1156.8** |
| BattleZone | 32850.4 | 30523.1 | 19369.3 | 35162.6 | 32625.4 | **35483.1** |
| BeamRider | 6044.1 | 5977.9 | 5197.7 | 5644.8 | 6070.2 | **6117.3** |
| Berzerk | 746.4 | 709.1 | 788.6 | 728.7 | 715.5 | **894.2** |
| Bowling | 36.0 | 32.5 | 45.5 | **61.9** | 31.5 | 29.9 |
| Boxing | 97.5 | 97.6 | 96.6 | 96.9 | 97.7 | **98.7** |
| Breakout | 56.9 | 59.8 | 57.7 | 58.1 | 63.2 | **75.8** |
| Carnival | 4805.4 | 4743.3 | **5040.4** | 4427.9 | 4623.5 | 4480.7 |
| Centipede | 4460.7 | 4909.7 | 4701.9 | 4613.3 | 4700.2 | **5154.6** |
| ChopperCommand | **9812.9** | 2306.4 | 9742.8 | 8967.3 | 9238.2 | 7611.4 |
| CrazyClimber | 135531.2 | 139641.8 | 133805.3 | 139011.0 | 138967.9 | **145380.2** |
| DemonAttack | 7570.5 | 6883.6 | 6268.4 | 5736.4 | 7298.5 | **26478.1** |
| DoubleDunk | 21.6 | 20.1 | -1.4 | **21.8** | 21.6 | -1.3 |
| ElevatorAction | 74358.0 | **74846.0** | 0.0 | 71432.9 | 73506.0 | 64614.0 |
| Enduro | 2155.8 | 2159.9 | 2132.2 | 2112.2 | 237.6 | **2271.2** |
| FishingDerby | **44.3** | 40.1 | 34.2 | 42.4 | 42.9 | 44.0 |
| Freeway | 33.6 | 33.5 | 33.5 | 33.6 | 33.6 | **33.8** |
| Frostbite | 7139.6 | 6980.3 | **7758.2** | 6777.8 | 5901.5 | 7391.7 |
| Gopher | 7815.1 | 7421.3 | 7295.3 | 7578.1 | 7581.9 | **12069.6** |
| Gravitar | **1763.4** | 1171.2 | 1050.4 | 1132.8 | 1165.9 | 1149.1 |
| Hero | 38102.2 | **38727.3** | 35444.2 | 37496.3 | 38217.7 | 35595.3 |
| IceHockey | 0.4 | 0.7 | -0.5 | -0.2 | -0.3 | **5.2** |
| Jamesbond | 586.0 | 650.6 | 747.1 | **896.6** | 663.7 | 733.9 |
| JourneyEscape | -1224.9 | -1261.3 | -3441.8 | -1241.1 | -1250.5 | **-992.9** |
| Kangaroo | 12676.2 | 12404.3 | **13028.7** | 12691.8 | 12834.1 | 12617.5 |
| Krull | 4250.4 | 4016.2 | 4171.0 | 4242.7 | 4046.5 | **4290.8** |
| KungFuMaster | 26459.1 | 23429.2 | 25450.0 | 27472.8 | 26899.8 | **34510.8** |
| MontezumaRevenge | 780.0 | **5188.8** | 5118.8 | 4364.6 | 2449.8 | 2486.2 |
| MsPacman | 3774.2 | 3749.3 | 3577.0 | 3694.1 | 3694.0 | **4376.9** |
| NameThisGame | 8309.9 | 8086.0 | 7598.0 | 7947.5 | 7810.1 | **11602.0** |
| Phoenix | 6340.5 | 7073.2 | 4779.7 | 7576.2 | 6902.3 | **14687.3** |
| Pitfall | -9.8 | **-1.6** | -5.4 | -15.1 | -18.8 | -2.0 |
| Pong | **20.1** | 19.4 | 20.1 | 19.6 | 19.6 | 19.7 |
| Pooyan | 3920.6 | 4125.0 | 3946.0 | 4103.6 | 3871.5 | **4731.5** |
| PrivateEye | 12284.7 | 3371.9 | 10081.2 | **26765.1** | 13566.5 | 216.6 |
| Qbert | 17261.8 | **19027.1** | 17926.5 | 17453.1 | 17072.6 | 18169.5 |
| Riverraid | 18317.0 | **19032.5** | 18281.6 | 18392.1 | 17705.1 | 13969.1 |
| RoadRunner | 52548.1 | 52626.6 | 52641.4 | **52853.6** | 52164.1 | 52048.7 |
| Robotank | 62.4 | 62.1 | 13.8 | 59.3 | 61.4 | **67.0** |
| Seaquest | 9605.5 | 7083.1 | **23277.5** | 6833.4 | 6403.1 | 9315.5 |
| Skiing | -25020.7 | -24296.5 | -23602.8 | **-23077.5** | -24954.3 | -25814.9 |
| Solaris | **2187.8** | 1877.9 | 1951.8 | 1375.9 | 1939.5 | 2032.2 |
| SpaceInvaders | 2621.1 | 2417.1 | 1682.1 | 2955.8 | 2604.4 | **5527.1** |
| StarGunner | 45718.0 | 46682.3 | 46281.6 | 48348.0 | 51118.0 | **55886.7** |
| Tennis | -0.8 | -0.1 | **22.8** | 22.7 | -1.1 | -0.0 |
| TimePilot | 10484.4 | 10637.3 | 5421.0 | 10474.8 | 10281.4 | **11161.8** |
| Tutankham | **240.8** | 232.4 | 147.8 | 240.2 | 223.3 | 212.0 |
| UpNDown | 21518.4 | 15871.6 | 11277.8 | 16540.0 | 17411.3 | **36217.2** |
| Venture | **1541.0** | 1515.0 | 1303.8 | 1470.2 | 1529.8 | 1528.2 |
| VideoPinball | **453118.7** | 431034.8 | 410651.4 | 431198.8 | 414069.2 | 327080.8 |
| WizardOfWor | 7172.8 | 6163.0 | 4719.3 | 6580.8 | 7323.6 | **8578.7** |
| YarsRevenge | 36432.8 | 42317.0 | 35471.4 | 37539.3 | 36775.4 | **44978.0** |
| Zaxxon | 14106.7 | 13750.8 | 8881.5 | 11179.1 | 14333.5 | **18290.2** |

Table 5: Raw scores for all games after 200M training frames

| Games | $\epsilon$-greedy | RND | CTS | PixelCNN | ICM | NoisyNets |
|---|---|---|---|---|---|---|
| AirRaid | 12719.7 | 12035.0 | 12296.9 | 12849.5 | **12923.5** | 10097.4 |
| Alien | 3461.2 | 3393.2 | 3177.4 | 3622.7 | 3455.5 | **4562.1** |
| Amidar | 2411.3 | 2263.8 | 2311.2 | 2499.6 | 2601.6 | **2837.4** |
| Assault | 3215.9 | 3247.1 | 2970.2 | 3026.7 | 1829.1 | **4929.5** |
| Asterix | 17337.1 | 18772.5 | 14855.6 | 19092.0 | 17287.0 | **42993.4** |
| Asteroids | 1609.5 | 1456.2 | 1350.5 | 1474.3 | 1442.0 | **3392.9** |
| Atlantis | 843635.3 | 796360.0 | 752722.0 | **874296.2** | 742868.0 | 860326.0 |
| BankHeist | 1041.1 | 1132.0 | 1086.1 | 621.9 | 1137.3 | **1255.0** |
| BattleZone | **38978.1** | 32521.0 | 29587.0 | 38601.4 | 37003.4 | 37314.9 |
| BeamRider | 5791.0 | 6416.3 | 5433.5 | 4604.3 | **6499.4** | 6016.0 |
| Berzerk | 772.9 | 772.1 | 845.5 | 779.3 | 740.5 | **938.4** |
| Bowling | 39.7 | 34.4 | 41.3 | **49.1** | 42.2 | 33.2 |
| Boxing | 98.2 | 98.4 | 97.8 | 98.2 | 98.5 | **98.9** |
| Breakout | 102.3 | 118.8 | 112.8 | 104.6 | 113.7 | **129.2** |
| Carnival | 4792.3 | 4726.3 | **5058.3** | 4385.5 | 4585.1 | 4464.1 |
| Centipede | 6470.3 | **6933.9** | 6572.6 | 6434.8 | 6600.6 | 6466.8 |
| ChopperCommand | **12578.4** | 2674.9 | 12133.2 | 12396.4 | 11595.2 | 9556.3 |
| CrazyClimber | 145511.7 | 146489.0 | 145465.8 | 147683.8 | 145570.3 | **153267.5** |
| DemonAttack | 19737.1 | 14349.2 | 11689.9 | 9350.5 | 18477.4 | **48634.6** |
| DoubleDunk | 22.3 | 21.9 | 10.3 | 22.1 | **22.3** | -0.2 |
| ElevatorAction | 75946.0 | 76118.0 | 0.0 | 67752.0 | 75606.0 | **81142.0** |
| Enduro | 2153.0 | 2175.9 | 2184.7 | 2170.9 | 31.6 | **2263.7** |
| FishingDerby | 46.4 | 41.4 | 44.3 | 44.7 | **47.5** | 45.6 |
| Freeway | 33.6 | 33.7 | 33.6 | 33.6 | 33.6 | **33.9** |
| Frostbite | 8048.7 | 8133.9 | 8343.3 | 7859.6 | 6909.5 | **9012.5** |
| Gopher | 10709.6 | 9630.3 | 9844.3 | 10547.0 | 10551.9 | **13382.3** |
| Gravitar | **1999.3** | 1245.6 | 1442.2 | 1338.2 | 1332.6 | 1337.8 |
| Hero | 43671.9 | **47802.0** | 41244.6 | 44286.2 | 45974.3 | 40075.3 |
| IceHockey | -1.0 | -1.0 | 2.7 | -0.9 | -1.4 | **13.6** |
| Jamesbond | 721.6 | 839.5 | 1094.4 | **1168.6** | 935.1 | 828.4 |
| JourneyEscape | -1185.4 | -1215.0 | -1992.6 | -1112.0 | -1199.6 | **-986.2** |
| Kangaroo | 12917.7 | 12606.1 | 12521.5 | 12611.1 | 12593.4 | **13981.5** |
| Krull | 4249.5 | 4202.2 | 4153.0 | 4075.4 | 4276.6 | **4371.8** |
| KungFuMaster | 28741.8 | 24087.6 | 26439.4 | 27234.6 | 28168.3 | **39382.1** |
| MontezumaRevenge | 780.0 | **5199.2** | 5028.2 | 4652.3 | 2795.9 | 2435.4 |
| MsPacman | 4063.2 | 4008.6 | 3964.5 | 3993.7 | 3861.0 | **5051.5** |
| NameThisGame | 9496.4 | 9202.3 | 8779.3 | 9147.4 | 9247.3 | **11499.7** |
| Phoenix | 6672.1 | 7318.6 | 4762.9 | 7671.8 | 8306.2 | **17579.5** |
| Pitfall | -5.3 | **-1.8** | -6.5 | -11.4 | -35.9 | -15.7 |
| Pong | **20.3** | 20.1 | 20.2 | 20.3 | 19.9 | 19.9 |
| Pooyan | 4324.6 | 4353.1 | 4416.6 | 4382.3 | 4128.3 | **4974.4** |
| PrivateEye | 26038.5 | 19421.7 | 5276.8 | 12211.4 | **26363.8** | 4339.9 |
| Qbert | 17606.8 | 18369.7 | 17247.3 | 17293.6 | 18603.9 | **18785.1** |
| Riverraid | 20607.1 | **20941.0** | 20694.7 | 20699.5 | 20290.8 | 16250.6 |
| RoadRunner | 54381.9 | 53829.7 | 56352.5 | 54773.0 | 53630.3 | **62423.5** |
| Robotank | 64.7 | 66.1 | 34.3 | 60.3 | 61.0 | **70.0** |
| Seaquest | 16044.8 | 11821.2 | **27027.8** | 11185.7 | 15469.9 | 20898.3 |
| Skiing | **-24988.0** | -28678.5 | -27801.4 | -29094.5 | -28781.4 | -29613.9 |
| Solaris | 1792.5 | 1684.3 | 1622.6 | 1252.0 | 1470.1 | **3024.6** |
| SpaceInvaders | 4172.0 | 3232.1 | 2229.4 | 4217.7 | 4936.0 | **9719.3** |
| StarGunner | 55112.9 | 57334.7 | 58005.6 | 56811.3 | 59500.1 | **63232.9** |
| Tennis | -0.1 | -0.6 | 23.0 | **23.2** | 2.6 | -0.0 |
| TimePilot | 11640.6 | 12010.6 | 7430.4 | 11564.4 | 9210.8 | **13593.5** |
| Tutankham | **239.4** | 237.9 | 193.0 | 234.7 | 233.6 | 238.1 |
| UpNDown | 38975.8 | 22559.4 | 14958.5 | 21799.9 | 24751.1 | **43345.9** |
| Venture | 1591.7 | 1554.7 | 1356.3 | 1536.6 | 1555.8 | **1646.4** |
| VideoPinball | 382581.4 | 447711.5 | 463181.6 | 454007.7 | 432640.6 | **478029.4** |
| WizardOfWor | 7939.4 | 6365.5 | 5807.5 | 7330.5 | 7041.1 | **10980.4** |
| YarsRevenge | 44142.9 | **56794.0** | 43173.2 | 47749.1 | 45821.0 | 52570.1 |
| Zaxxon | 15101.4 | 14532.9 | 10559.9 | 12248.3 | 8451.7 | **19211.5** |

