# OpenReview forum: "On Bonus Based Exploration Methods In The Arcade Learning Environment"
_ICLR.cc/2020/Conference — Accept (Poster)_

### Official Review · AnonReviewer1 · 2019-10-22
**Official Blind Review #1**

**Rating:** 6

**Review:**

Updated review: I am overall happy with the response of the authors. I can appreciate the contributions of the paper and I am happy to recommend accept. The empirical study offers some insights into deep RL methods for ATARI games and raises some key questions. I feel the current version of the paper does not build upon these insights to propose a new method.

-------------------------------------------------------------------------------------------------------------------------------------------------------------------------

Summary: This paper presents a detailed empirical study of the recent bonus based exploration method on the Atari game suite. The paper concludes that methods that perform well on Montezuma’s revenge do not necessarily perform well on the other games, sometimes, even worse than the eps-greedy approach. This also leads to the conclusion that recent results on the game Montezuma’s revenge can be attributed to architectural changes instead of the exploration method.

I think this is a-ok paper in that it does what it says it does. The paper is clear and well-written.

I think the main contribution of the paper is that it raises some questions over existing methods/trends in solving exploration problems in reinforcement learning by comparing the performance of multiple methods across various games in ATARI suite.
I think this is relevant to the ICLR community and will be appreciated by it.

However, I also feel that while the paper runs a satisfactory empirical analysis, it was all too much focussed on the existing methods. Throughout the paper, the experiments and results raise questions on the robustness and generalization of existing exploration methods across various ATARI games, but the paper puts absolutely zero effort into investigating if there is a quick fix to the questions it poses. For example, one could easily investigate in the CTS method if the factor by which exploration bonus dies N^{alpha} (alpha=-1/2 by default) changes, then does it do better or worse (more below on this).
I can understand that might not be the aim of the paper but still.

Here are a couple of points that I felt conflicted/confused about the paper:
- The conclusion of the paper is that ‘progress of exploration in ATARI suite is obfuscated by good results in single domain’. I am confused if the paper is making a narrow point that (1) dont focus on Montezuma’s revenge OR (2) is it admitting a broader point that focussing on even ATARI is probably not a good choice. I am not saying that I know the answer to this question, but I am unclear as to what is the question the paper is trying to raise. If it is saying (1st) then I find it contradictory that it is not ok to focus on MR but it is ok to focus on ATARI as a single domain; if it is saying the second then also it is contradictory because the paper only experiments with the ATARI suite.

- It is interesting to note that noisy networks are most robust to hyperparameter optimization on a separate set of games when tested on a different set of games. It is also interesting to note that noisy networks are the only exploration bonus method that does not decrease/reduce exploration as the experience of the agent increases. I would have liked to see if the paper had made an attempt to investigate this. I feel such a hypothesis would have been easy to investigate with simple modifications to the CTS methods. Currently, the exploration bonus goes down by the factor of 1/sqrt(N)  in the CTS method. A comparison that showed the performance of CTS for a couple more values of factors such as (1/N) or (1/N)^{1/4} would have been nice to see if that mattered.

- One of the comparisons I did not particularly find fair was when the hyperparameters of various methods were tuned to play MR and then the hyperparameters were fixed and the method were tested on other ATARI games.

- Another point I felt was missing was checking if rainbow DQN is really the reason behind the observed performance of the methods. It would have been interesting to know how the methods performed when combined with the original DQN algorithm.


**Experience Assessment:**

I have read many papers in this area.

**Review Assessment: Checking Correctness Of Derivations And Theory:**

I assessed the sensibility of the derivations and theory.

**Review Assessment: Checking Correctness Of Experiments:**

I assessed the sensibility of the experiments.

**Review Assessment: Thoroughness In Paper Reading:**

I read the paper at least twice and used my best judgement in assessing the paper.

---

> ### Author Response · Authors · 2019-11-09
> **Response to Reviewer 1**
>
> Thank your feedback and taking the time to go through our manuscript.
>
> “This also leads to the conclusion that recent results on the game Montezuma’s revenge can be attributed to architectural changes instead of the exploration method.”
>
> We do not disagree that progress in MR can be attributed to the exploration method. However, we argue that the benefits of these methods do not translate to other games in the ALE.
>
> “the paper puts absolutely zero effort into investigating if there is a quick fix to the questions it poses”
>
> It is true that we focused primarily on current practices in exploration research. It seemed important to us to highlight how these practices have impacted the field of exploration in RL. Moreover, we wanted to know how we may have been misled regarding our progress in exploration.
>
> Given the experiments we performed it is not clear whether a "quick fix" exists. It seems more likely that new methods will have to be designed, which take our findings into account.
>
> “A comparison that showed the performance of CTS for a couple more values of factors such as (1/N) or (1/N)^{1/4} would have been nice to see if that mattered.”
>
> We thought this direction did not hold much promise and thus we did not investigate further. There are theoretical reasons for this particular value choice (see [1,  2]). 1/N with DQN has been done in [3] in Figure 10 and led to a significant performance drop.
>
> [1] An Analysis of Model-Based Interval Estimation for Markov Decision Processes, Strehl and Littman (2006).
> [2] Near-Bayesian exploration in polynomial time, Kolter and Ng (2009).
> [3] Unifying Count-Based Exploration and Intrinsic Motivation, Bellemare et al. (2016)
>
> “If it is saying (1st) then I find it contradictory that it is not ok to focus on MR but it is ok to focus on ATARI as a single domain;”
>
> It is (1). Focusing on the ALE as a single domain is in line with the current use of the ALE for research in reinforcement learning. Limiting oneself just to Montezuma’s Revenge or the hard exploration games is a practice that is unique to exploration practitioners.
> Moving beyond the ALE will be of interest in the future; however, our results show that efficient exploration in the ALE is currently far from being solved. We think that evaluating on a set of 60 diverse games is already a step up from only using 7 games.
>
> “It is interesting to note that noisy networks are most robust to hyperparameter optimization on a separate set of games when tested on a different set of games.”
>
> We did not tune NoisyNets and kept the original hyperparameters. We will update the paper to make this more clear.
>
> “It is also interesting to note that noisy networks are the only exploration bonus method that does not decrease/reduce exploration as the experience of the agent increases.”
>
> It is true that the amount of noise injected by NoisyNets is learned. NoisyNets are then able to modulate the amount of exploration during training. In contrast, the bonuses from other methods will shrink in states that have already been visited.
>
> “One of the comparisons I did not particularly find fair was when the hyperparameters of various methods were tuned to play MR and then the hyperparameters were fixed and the method were tested on other ATARI games.”
>
> What about the comparisons do you not find fair? If you find splitting the games in the ALE into a train and test set of games is unfair, then we would argue that this is a standard procedure (see [3,4]).
> The choice of games in the training set is open to debate and for this reason we evaluate its impact in Section 3.4.
> If it is something else you find unfair, please let us know.
>
> [3] The Arcade Learning Environment: An Evaluation Platform for General Agents, Bellemare et al. (2012)
> [4] Revisiting the Arcade Learning Environment: Evaluation Protocols and Open Problems for General Agents, Machado et al. (2017)
>
> “Another point I felt was missing was checking if rainbow DQN is really the reason behind the observed performance of the methods. It would have been interesting to know how the methods performed when combined with the original DQN algorithm.”
>
> We agree that a more detailed study would be helpful to explain why the gap between $\epsilon$-greedy and exploration bonuses has shrunk. We postulate that this is mostly due to the use of a prioritized replay buffer and we will add experiments without prioritization.

---

### Official Review · AnonReviewer3 · 2019-10-26
**Official Blind Review #3**

**Rating:** 6

**Review:**

#rebuttal responses

I change the score to be weak accept as the authors do not provide any comparison result on Rainbow without the prioritized replay buffer during the rebuttal phase. I also agree with Reviewer 1's opinion that the authors do not provide some fixing method, such as combining the noisy networks and bonus methods.


#review
This paper evaluates the recently proposed exploration methods that achieve ground-breaking performance in the difficult exploration problem, Montezuma's Revenge.  The authors combine Rainbow with different exploration methods, such as count-based bonus methods, curiosity-driven methods, and noisy networks. Results show that these methods fail to beat epsilon-greedy on other Atari games, even if the parameters of these methods are tuned.

The paper is very well written, and they claim that evaluating the exploration methods on the Montezuma's Revenge and tuning parameters on this environment are not suitable for the total ALE environments. The claim is very interesting and important for the exploration community.

To support their claim, the authors firstly compare bonus exploration methods, noisy networks, and epsilon-greedy on hard exploration games. Then results in easy games and other games are presented. The results are very impressive.

Question:
(1) The authors compared these methods based on Rainbow, which employs many techniques, such as the prioritized replay buffer. Can you show the comparison results on Rainbow without the prioritized replay buffer? It will strengthen the understanding of these exploration methods.
(2) The noisy networks perform well on most games, while bonus methods perform well on hard games. Is there any combination method to achieve better performance?

**Experience Assessment:**

I have read many papers in this area.

**Review Assessment: Checking Correctness Of Derivations And Theory:**

I carefully checked the derivations and theory.

**Review Assessment: Checking Correctness Of Experiments:**

I carefully checked the experiments.

**Review Assessment: Thoroughness In Paper Reading:**

I read the paper thoroughly.

---

> ### Author Response · Authors · 2019-11-09
> **Response to Reviewer 3**
>
> Thank you for your review and taking the time to read our paper.
>
>  “(1) Can you show the comparison results on Rainbow without the prioritized replay buffer? It will strengthen the understanding of these exploration methods.”
>
> We agree that these results would be helpful and we will add them in a future revision of the paper.
>
> (2) The noisy networks perform well on most games, while bonus methods perform well on hard games. Is there any combination method to achieve better performance?
>
> Bonus methods seem to perform well on Montezuma’s Revenge, but they do not perform well on the remaining hard exploration games. We think a naive combination of NoisyNets and exploration bonuses would likely combine their weaknesses. As such, it is not clear right now how to combine their benefits, so we leave it to future work.

---

### Decision · Program_Chairs · 2019-12-19

**Decision:**

Accept (Poster)

**Comment:**

This paper presents a detailed comparison of different bonus-based exploration methods on a common evaluation framework (Rainbow) when used with the ATARI game suite. They find that while these bonuses help on Montezuma's Revenge (MR), they underperform relative to epsilon-greedy on other games. This suggests that architectural changes may be a more important factor than bonus-based exploration in recent advances on MR.

The reviewers commented that this paper makes no effort to present new techniques, and the insights discovered could be expanded on. Despite this, it is an interesting paper that is generally well argued and would be a useful contribution to the field. I recommend acceptance.